# Criticality supports cross-frequency cortical-thalamic information transfer during conscious states

Daniel Toker[1,2]*, Eli Müller[3], Hiroyuki Miyamoto[4,5,6], Maurizio S Riga[7], Laia Lladó-Pelfort[8], Kazuhiro Yamakawa[4,9], Francesc Artigas[10,11,12], James M Shine[3], Andrew E Hudson[13,14], Nader Pouratian[15]†, Martin M Monti[2,16]†

[1]Department of Neurology, University of California, Los Angeles, Los Angeles, United States; [2]Department of Psychology, University of California, Los Angeles, Los Angeles, United States; [3]Brain and Mind Centre, University of Sydney, Sydney, Australia; [4]Laboratory for Neurogenetics, RIKEN Center for Brain Science, Saitama, Japan; [5]PRESTO, Japan Science and Technology Agency, Saitama, Japan; [6]International Research Center for Neurointelligence, University of Tokyo, Nagoya, Japan; [7]Andalusian Center for Molecular Biology and Regenerative Medicine, Seville, Spain; [8]Departament de Ciències Bàsiques, Universitat de Vic-Universitat Central de Catalunya, Barcelona, Spain; [9]Department of Neurodevelopmental Disorder Genetics, Institute of Brain Science, Nagoya City University Graduate School of Medical Science, Nagoya, Japan; [10]Departament de Neurociències i Terapèutica Experimental, CSIC-Institut d'Investigacions Biomèdiques de Barcelona, Barcelona, Spain; [11]Institut d'Investigacions Biomèdiques August Pi i Sunyer (IDIBAPS), Barcelona, Spain; [12]Centro de Investigación Biomédica en Red de Salud Mental (CIBERSAM), Instituto de Salud Carlos III, Madrid, Spain; [13]Department of Anesthesiology, Veterans Affairs Greater Los Angeles Healthcare System, Los Angeles, United States; [14]Department of Anesthesiology and Perioperative Medicine, University of California, Los Angeles, Los Angeles, United States; [15]Department of Neurological Surgery, UT Southwestern Medical Center, Dallas, United States; [16]Department of Neurosurgery, University of California, Los Angeles, Los Angeles, United States

*For correspondence:
danieltoker@g.ucla.edu

†co-senior authors

**Competing interest:** The authors declare that no competing interests exist.

**Abstract** Consciousness is thought to be regulated by bidirectional information transfer between the cortex and thalamus, but the nature of this bidirectional communication - and its possible disruption in unconsciousness - remains poorly understood. Here, we present two main findings elucidating mechanisms of corticothalamic information transfer during conscious states. First, we identify a highly preserved spectral channel of cortical-thalamic communication that is present during conscious states, but which is diminished during the loss of consciousness and enhanced during psychedelic states. Specifically, we show that in humans, mice, and rats, information sent from either the cortex or thalamus via δ/θ/α waves (~1–13 Hz) is consistently encoded by the other brain region by high γ waves (52–104 Hz); moreover, unconsciousness induced by propofol anesthesia or generalized spike-and-wave seizures diminishes this cross-frequency communication, whereas the psychedelic 5-methoxy-N,N-dimethyltryptamine (5-MeO-DMT) enhances this low-to-high frequency interregional communication. Second, we leverage numerical simulations and neural electrophysiology recordings from the thalamus and cortex of human patients, rats, and mice to show that these changes in cross-frequency cortical-thalamic information transfer may be mediated by excursions of low-frequency thalamocortical electrodynamics toward/away from edge-of-chaos criticality, or the phase transition from stability to chaos. Overall, our findings link thalamic-cortical communication

to consciousness, and further offer a novel, mathematically well-defined framework to explain the disruption to thalamic-cortical information transfer during unconscious states.

## Editor's evaluation

This important study investigates thalamocortical communication and cross-frequency coupling in human and animal models under anesthesia, seizures, and the effects of the serotonergic psyche-delic compound 5-MeO-DMT. These findings are exciting and compelling because they put different perturbations of brain functions – anesthesia, seizures, and psychedelic stimulation – into a single modeling framework demonstrating how these opposing perturbations reduce and enhance thalam-ocortical communication at specific frequencies. The evidence is compelling because it comes from multiple animal models and also incorporates a state-of-the-art neural mass model to investigate critical brain dynamics.

## Introduction

Mounting evidence suggests that the maintenance of cortical information processing during conscious states requires preserved communication between the cortex and several key subcortical structures (*Koch et al., 2016*). Among the subcortical structures that have been implicated in large-scale neural information processing during normal waking states, the thalamus stands out as perhaps the most important (*Shine, 2021*). This is most clearly suggested by its anatomy: the first-order nuclei of thal-amus are the major anatomical bridges across which sensory information is transferred from peripheral sources to the cortex, and the presence of extensive connections between higher order thalamic nuclei and diverse cortical regions suggests that these nuclei are among the key bridges through which infor-mation is transferred from one part of the cortex to another (*Sherman, 2007*; *Sherman, 2016*; *Shine, 2021*) - a hypothesis which has found support from diverse neuroimaging studies (*Saalmann et al., 2012*; *Theyel et al., 2010*; *Hwang et al., 2017*; *Müller et al., 2020*). It is therefore unsurprising that unconsciousness, which consistently coincides with disrupted cortical information processing (*Imas et al., 2005*; *Toker et al., 2022*; *Sanjari et al., 2021*; *Schroeder et al., 2016*; *Hudetz et al., 2020*; *Ku et al., 2011*; *Lee et al., 2013*; *Mäki-Marttunen et al., 2013*; *Chen et al., 2020*), also appears to consistently coincide with disrupted communication between the cortex and thalamus (*Zheng et al., 2017*; *White and Alkire, 2003*; *Malekmohammadi et al., 2019*; *Redinbaugh et al., 2020*; *Bastos et al., 2021*; *Afrasiabi et al., 2021*). Identifying the mechanisms supporting cortical-thalamic commu-nication, and how this communication may be disrupted during unconscious states, is therefore crucial both to our basic understanding of large-scale neural computation, as well as our clinical grasp on conditions in which cortical-subcortical communication appears to be disrupted, such as in coma and vegetative states (*Monti et al., 2010*).

One unexplored mechanism which may support bidirectional communication between the cortex and thalamus during conscious states is criticality. Criticality, or a critical point, refers to the transition between different phases of a system, such as different phases of matter (e.g. solid versus liquid) or different phases of temporal dynamics (e.g. asynchronous versus synchronous dynamics, or laminar versus turbulent airflow). It is by now well-established that critical and near-critical systems tend to have a high capacity for transmitting and encoding information (*Langton, 1990*; *Crutchfield and Young, 1988*; *Boedecker et al., 2012*; *Bertschinger and Natschläger, 2004*). There are many types of critical points, but in the context of neuroscience, two types of critical points, namely avalanche criticality and edge-of-chaos criticality, may be particularly relevant, as both been associated with the dynamics of the waking, healthy brain (*O'Byrne and Jerbi, 2022*). In this study, we focus exclusively on the edge-of-chaos transition, a critical point that is perhaps particularly relevant for supporting information processing in the brain (*O'Byrne and Jerbi, 2022*; *Toker et al., 2022*) and in complex systems more generally (*Langton, 1990*; *Crutchfield and Young, 1988*; *Boedecker et al., 2012*; *Bertschinger and Natschläger, 2004*). In our recent work (*Toker et al., 2022*), we showed that slow cortical electrodynamics during conscious states specifically operate near the edge-of-chaos critical point, or the transition between periodicity and chaos, and that this form of criticality supports the information-richness of waking cortical electrodynamics. We also showed that slow cortical electro-dynamics transition away from this critical point during anesthesia, generalized seizures, and coma

(which diminishes the information-richness of cortical activity), and that slow cortical electrodynamics transition closer to this critical point following the administration of the serotonergic hallucinogen lysergic acid diethylamide (which enhances the information-richness of cortical activity; *Toker et al., 2022*). These results accord with the broad empirical evidence suggesting that cortical activity transitions away from criticality during unconscious states and transitions closer to criticality during psychedelic states (*Zimmern, 2020*). Therefore, it is straightforward to predict that the proximity of slow neural electrodynamics to the edge-of-chaos critical point might similarly modulate the strength of bidirectional communication between the cortex and thalamus during normal waking, unconscious, and psychedelic states (*Figure 1*).

Here, in order to better characterize the mechanisms of cortical-thalamic communication and how those mechanisms might be modulated by the proximity of neural electrodynamics to edge-of-chaos criticality, we first applied a novel information-theoretic measure of spectrally resolved information transfer to concurrent thalamic and cortical electric field recordings across species, including human essential tremor (ET) patients, Long-Evans rats, Genetic Absence Epilepsy Rats from Strasbourg (GAERS rats), and C57BL/6 mice. We identified a highly preserved pattern of low-to-high frequency bidirectional cortical-thalamic information transfer present across nearly all patients and animals during conscious states. Specifically, we found that information transmitted at low frequencies (~1–13 Hz) from one brain structure is consistently encoded by the other brain structure at high frequencies (52–104 Hz). We also present evidence that this cross-frequency cortical-thalamic information transfer is disrupted during unconsciousness induced by both $\gamma$-Aminobutyric acid mediated (GABAergic) anesthetics and generalized spike-and-wave seizures, and enhanced by the serotonergic hallucinogen 5-methoxy-*N,N*-dimethyltryptamine, or 5-MeO-DMT, a potent dual agonist of 5-HT$_{1A}$ and 5-HT$_{2A}$ receptors. Finally, drawing both on our analysis of our electrophysiology recordings and on numerical simulations using a novel mean-field model of the basal ganglia-thalamo-cortical system, we found that the strength of this cross-frequency cortical-thalamic information transfer across brain

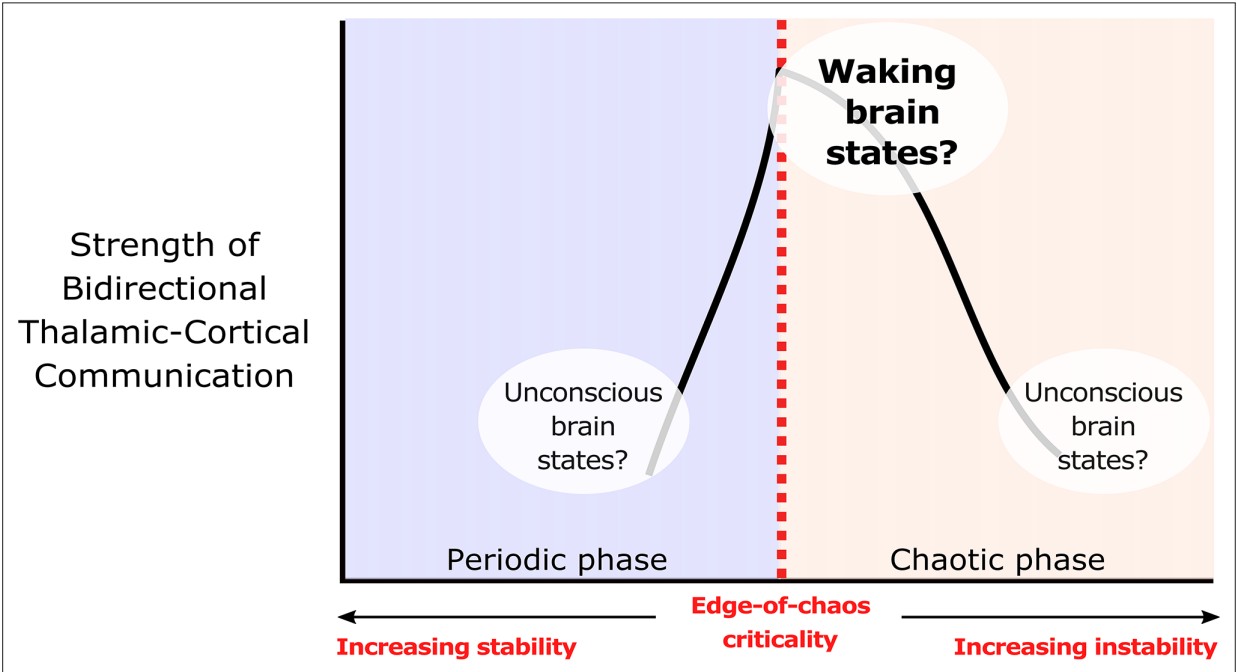

**Figure 1.** We hypothesize that edge-of-chaos criticality supports thalamic-cortical communication during waking brain states. We hypothesize that the strength of bidirectional information transfer between the cortex and thalamus should be highest during waking brain states, owing to the proximity of slow neural electrodynamics to edge-of-chaos criticality during these states. We also predict that as slow neural electrodynamics transition away from this critical point during unconscious states, either into the chaotic phase or into the periodic phase, the strength of cortical-thalamic information transfer should be diminished.

states is likely mediated by transitions of low-frequency thalamocortical electrodynamics toward or away from edge-of-chaos criticality, as predicted. To our knowledge, this work is the first to show that this precise form of criticality supports interregional communication in the brain.

## Results

### Low-to-high-frequency information transfer between the thalamus and cortex is highly preserved across humans, rats, and mice in waking states

Because long-range neural communication is likely frequency-multiplexed, with distinct long-range information streams encoded by distinct (and interacting) frequencies of oscillatry neural electrodynamics (*Akam and Kullmann, 2014*; *Panzeri et al., 2010*; *Chao et al., 2018*; *Fontolan et al., 2014*; *Malekmohammadi et al., 2015*), we first evaluated patterns of thalamic-cortical communication during conscious states using a recently developed, spectrally resolved measure of directed information transfer which is both model-free and sensitive to delayed interactions (*Pinzuti et al., 2020*). The measure evaluates the strength and significance of frequency-specific information transfer using surrogate time-series, which enable the estimation of how many bits of transfer entropy are lost when dynamics only within certain frequency ranges are randomized (see Materials and methods). We applied this spectral information transfer measure to neural extracellular electric fields recorded simultaneously during waking states from the ventral intermediate (Vim) thalamic nucleus and ipsilateral sensorimotor cortex of human essential (ET) patients, and from the ventral posterior thalamic nucleus and contralateral somatosensory cortex of GAERS rats. For the GAERS rats, waking states were strictly separated from spike-and-wave seizure periods (as assessed through visual inspection of the data), ensuring that waking state data for these animals were free from epileptic activity. We also analyzed information transfer between the ventral posterior thalamic nucleus and ipsilateral somatosensory cortex of Long-Evans rats, as well as the mediodorsal thalamic nucleus and the ipsilateral medial prefrontal cortex of C57BL/6 mice. The inclusion of these normal, healthy animals provided a crucial control for the study, by helping to rule out the possibility that any observed patterns of spectral information transfer in the human ET patients and GAERS rats are driven by pathological brain activity. Note that with the exception of the recording locations in the GAERS rats, all of these thalamic nuclei share direct reciprocal anatomical connections with the cortical areas from which signals were simultaneously recorded. Although the recording sites in the GAERS rats are not directly connected, the ventral posterior thalamic nucleus communicates indirectly with the contralateral somatosensory cortex via its reciprocal connectivity with the ipsilateral somatosensory cortex, which directly projects to the contralateral somatosesory cortex (*Petreanu et al., 2007*; *Wise and Jones, 1976*; *Olavarria et al., 1984*).

Using one half of all patients'/animals' 10 s trials, we first performed an exploratory sweep of all possible spectral patterns of information transfer between the cortex and thalamus across all patients/animals, channels, and recording windows. Because of the prohibitive computational cost of evaluating information transfer across all possible pairs of frequency bands using a large number of surrogates, we only used five surrogate time-series (per 10 s trial) for this exploratory sweep. By doing so, we identified a possible spectral channel of cortical-thalamic communication present across all evaluated species during conscious states (*Figure 2*): namely, information sent from either the cortex or thalamus in the low-frequency range (~1–13 Hz) seemed to be consistently encoded by the other brain region in the high $\gamma$ range (52–104 Hz) (note that these exact frequency ranges are determined by successive halves of the sampling frequency, as this method is based on wavelet decomposition - see Materials and methods). To confirm this finding, we re-ran this spectral information transfer analysis on the remaining half of each patient's and animal's 10 s trials, along just these frequency bands, but using sufficient surrogates (100) to evaluate statistical significance. We found that there was indeed significant low-to-high frequency bidirectional cortical-thalamic information transfer across nearly all subjects during conscious states (*Table 1*). To validate the pervasiveness of this phenomenon across subjects, we performed a binomial test treating each subject as a Bernoulli trial with a success being a significant harmonic mean p-value less than 0.05. Under the null hypothesis (i.e., assuming the event of a significant p-value for a subject is a random occurrence with a probability of 0.05), this test returned p=0 for both cortico-thalamic and thalamo-cortical cross-frequency information transfer,

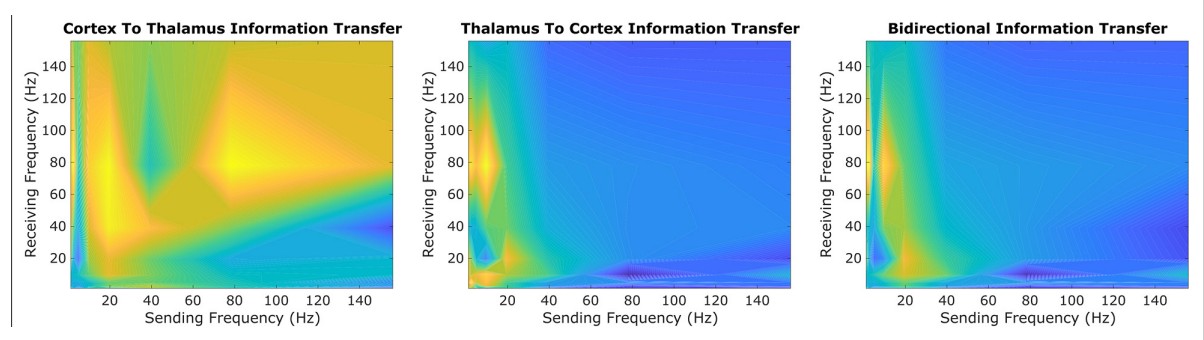

**Figure 2.** An exploratory sweep suggests that thalamus and cortex transmit information bidirectionally from low-to-high frequencies during conscious states. In our initial exploratory sweep of spectral patterns of directed cortical-thalamic information transfer during conscious states, based on half of all patients'/animals' trials, we identified a prominent motif of low-to-high frequency bidirectional communication that was present during waking states in nearly all subjects and species. We first estimated the (z-scored) strengths of information transfer across every possible pair of frequency bands, for every 10 s trial, and for every subject during waking states. We then took the average cross-trial result for every subject. Here, we plotted the mode across subjects' cross-trial averages in order to reveal the spectral patterns of information transfer that occurred most frequently across subjects during conscious states. For cortico-thalamic information transfer (left), we found that information sent from the cortex across all frequencies is frequently received by the thalamus in the high γ range. For thalamo-cortical information transfer (middle), we observed a prominent pattern of low-to-high frequency information transfer. When looking at the mode across all cross-trial averages of both cortico-thalamic and thalamo-cortical information transfer during conscious states (right), there seems to be a consistent channel of communication from the low-frequency range (~1–13Hz) to the high-frequency range (52–104 Hz) in both directions (cortico-thalamic and thalamo-cortical). We therefore chose to study this cross-frequency pattern of information transfer in our subsequent analyses of waking, GABAergic anesthesia, generalized spike-and-wave seizure, and psychedelic states.

confirming that the observed cross-frequency information transfer is a common feature of conscious states across the mammals studied. To further substantiate this finding, we conducted an analysis on a smaller subset of our data using a larger number of surrogates (250) and again found statistically significant low-to-high frequency information transfer between thalamus and cortex during conscious states (*Supplementary file 1*).

## Bidirectional cross-frequency cortical-thalamic information transfer is disrupted in unconsciosuness and enhanced during psychedelic states

To test whether this low-to-high frequency cortical-thalamic communication is disrupted during unconscious states and enhanced during psychedelic states (see Introduction), we calculated the strength of low-to-high-frequency bidirectional information transfer following intravenous administration of propofol anesthesia in human ET patients (varying doses - see Materials and methods) and Long-Evans rats (plasma propofol concentration of 12 μg/ml; Long-Evans rats were included so as to rule out the possibility that observed effects of anesthesia in the ET patients were driven by their pathology); during spontaneous generalized spike-and-wave seizures in GAERS rats; and following subcutaneous injection of saline +5-MeO-DMT (5 mg/kg) in C57BL/6 mice. For these analyses, all trials were used (rather than half of the 10 s windows of data, as we did for the waking state data in *Figure 2* and *Table 1*). As predicted, we found that cross-frequency information transfer from the cortex to the thalamus was disrupted during unconscious states and enhanced during psychedelic states. Specifically, propofol diminished low-frequency to high-frequency information transfer from the cortex to the thalamus in both human ET patients (p=0.002, one-tailed Wilcoxon signed-rank test comparing patients' cross-trial medians during waking states versus propofol states) (*Figure 3A*) and Long-Evans rats (p=0.002; *Figure 3B*). Similarly, cross-frequency corticothalamic information transfer was reduced during generalized spike-and-wave seizures in GAERS rats (p=0.0078; *Figure 3C*). Conversely, 5-MeO-DMT significantly increased the strength of low-to-high frequency corticothalamic information transfer in C57BL/6 mice (p=0.0312; *Figure 3D*), despite the fact that this brain state, similar to anesthesia, is marked by reduced high-frequency activity and increased low-frequency activity in both thalamus and cortex (*Figure 4*); this suggests that these observed changes to cross-frequency communication are independent of the spectral content of thalamocortical electrodynamics. The same overall pattern was seen with low-to-high frequency information transfer from the thalamus to the cortex. Specifically, we found that the strength cross-frequency communication from

**Table 1.** Statistical analysis confirms that thalamus and cortex transmit information bidirectionally from low-to-high frequencies during waking states.

Following our initial exploratory sweep of all possible spectral patterns of cortical-thalamic communication (*Figure 2*), which was based on one half of the 10 s trials for each patient/animal, we used surrogate testing on the remaining half of trials to evaluate whether there was statistically significant information transfer from slow (~1–13 Hz) to fast (52–104 Hz) electrodynamics between anatomically connected sub-regions of the thalamus and cortex (see Materials and methods). For each 10 s window of activity, surrogate testing produced a single p-value reflecting the significance of cross-frequency information transfer in each direction (cortico-thalamic and thalamo-cortical). Overall statistical significance, across 10 s windows within each subject, was assessed by evaluating the harmonic mean $\mathring{p}$ (*Wilson, 2019*) of all single-trial p-values. The number of 10 s trials used in this analysis for each patient/animal are listed here in the right-hand column. In line with our initial exploratory sweep (*Figure 2*), we found that there was significant low-to-high frequency bidirectional information transfer between the thalamus and cortex in nearly every species, strain, and subject. Note that we did not correct for multiple comparisons because each subject's harmonic mean p-value contributes to a single overarching hypothesis about inter-region neural communication, rather than representing separate, independent hypotheses.

| | Cortex to Thalamus | Thalamus to Cortex | Number of Trials |
|---|---|---|---|
| Human ET Patient 1 | $\mathring{p} = 0.0235$ | $\mathring{p} = 0.0099$ | 3 |
| Human ET Patient 2 | $\mathring{p} = 0.0099$ | $\mathring{p} = 0.0099$ | 3 |
| Human ET Patient 3 | $\mathring{p} = 0.0099$ | $\mathring{p} = 0.0099$ | 3 |
| Human ET Patient 4 | $\mathring{p} = 0.0099$ | $\mathring{p} = 0.0099$ | 2 |
| Human ET Patient 5 | $\mathring{p} = 0.0099$ | $\mathring{p} = 0.0099$ | 3 |
| Human ET Patient 6 | $\mathring{p} = 0.0099$ | $\mathring{p} = 0.0099$ | 2 |
| Human ET Patient 7 | $\mathring{p} = 0.0099$ | $\mathring{p} = 0.0099$ | 3 |
| Human ET Patient 8 | $\mathring{p} = 0.0099$ | $\mathring{p} = 0.0099$ | 1 |
| Human ET Patient 9 | $\mathring{p} = 0.0099$ | $\mathring{p} = 0.0099$ | 3 |
| Human ET Patient 10 | $\mathring{p} = 0.0099$ | $\mathring{p} = 0.0099$ | 3 |
| Long-Evans Rat 1 | $\mathring{p} = 0.043$ | $\mathring{p} = 0.024$ | 6 |
| Long-Evans Rat 2 | $\mathring{p} = 0.0185$ | $\mathring{p} = 0.0394$ | 6 |
| Long-Evans Rat 3 | $\mathring{p} = 0.0547$ | $\mathring{p} = 0.0394$ | 7 |
| Long-Evans Rat 4 | $\mathring{p} = 0.0343$ | $\mathring{p} = 0.0319$ | 7 |
| Long-Evans Rat 5 | $\mathring{p} = 0.0499$ | $\mathring{p} = 0.0378$ | 6 |
| Long-Evans Rat 6 | $\mathring{p} = 0.0317$ | $\mathring{p} = 0.0659$ | 6 |
| Long-Evans Rat 7 | $\mathring{p} = 0.0226$ | $\mathring{p} = 0.0744$ | 5 |
| Long-Evans Rat 8 | $\mathring{p} = 0.0347$ | $\mathring{p} = 0.0173$ | 1 |
| Long-Evans Rat 9 | $\mathring{p} = 0.0792$ | $\mathring{p} = 0.1683$ | 3 |
| GAERS Rat 1 | $\mathring{p} = 0.0385$ | $\mathring{p} = 0.0265$ | 71 |
| GAERS Rat 2 | $\mathring{p} = 0.0307$ | $\mathring{p} = 0.0193$ | 31 |
| GAERS Rat 3 | $\mathring{p} = 0.033$ | $\mathring{p} = 0.0257$ | 159 |
| GAERS Rat 4 | $\mathring{p} = 0.0319$ | $\mathring{p} = 0.0291$ | 148 |

*Table 1 continued on next page*

Table 1 continued

|  | Cortex to Thalamus | Thalamus to Cortex | Number of Trials |
|---|---|---|---|
| GAERS Rat 5 | $\mathring{p} = 0.0391$ | $\mathring{p} = 0.0295$ | 80 |
| GAERS Rat 6 | $\mathring{p} = 0.0584$ | $\mathring{p} = 0.0325$ | 55 |
| GAERS Rat 7 | $\mathring{p} = 0.0338$ | $\mathring{p} = 0.029$ | 79 |
| C57BL/6 Mouse 1 | $\mathring{p} = 0.0268$ | $\mathring{p} = 0.0238$ | 104 |
| C57BL/6 Mouse 2 | $\mathring{p} = 0.0247$ | $\mathring{p} = 0.0215$ | 78 |
| C57BL/6 Mouse 3 | $\mathring{p} = 0.0278$ | $\mathring{p} = 0.0224$ | 76 |
| C57BL/6 Mouse 4 | $\mathring{p} = 0.0425$ | $\mathring{p} = 0.0349$ | 76 |
| C57BL/6 Mouse 5 | $\mathring{p} = 0.0421$ | $\mathring{p} = 0.0331$ | 81 |

the thalamus to the cortex was significantly diminished during propofol anesthesia in both human ET patients (p=0.002; *Figure 5A*) and Long-Evans rats (p=0.0098; *Figure 5B*). Similarly, the strength of cross-frequency thalamocortical information transfer was significantly reduced in GAERS rats during generalized spike-and-wave seizures (p=0.0078; *Figure 5C*), but did not change during psychedelic states in C57BL/6 mice (p=0.3125; *Figure 5D*).

To confirm that the observed results reflect a breakdown in thalamic-cortical communication rather than changes in the spectral content of thalamocortical electrodynamics, we performed a permutation-based nonparametric analysis of covariance, which revealed significant variance across brain states in the strength of both cross-frequency cortico-thalamic (p=0.0001) and thalamo-cortical (p=0.0001) information transfer, which could not be explained by spectral changes at either low (1–13 Hz) or high (52–104 Hz) frequencies (*Supplementary file 2*). We also confirmed that these observed changes to cross-frequency communication were not driven by changes in non-spectrally resolved information transfer between the thalamus and cortex. Specifically, we found that (non-spectrally resolved) transfer entropy between these two brain regions did not vary consistently across different brain states, instead decreasing significantly during unconsciousness only in human ET patients, and increasing significantly during propofol anesthesia in Long-Evans Rats, generalized spike-and-wave seizures in GAERS rats, and psychedelic states in C57BL/6 mice from both cortex to thalamus (*Figure 3—figure supplement 1*) and thalamus to cortex (*Figure 5—figure supplement 1*). We also found that the observed results were not driven by changes to the strength of phase-amplitude coupling between these regions. Specifically, we found that coupling between the phase of low-frequency (1–13 Hz) activity in one brain region and the amplitude of high-frequency (52–104 Hz) activity in the other, as assessed using the Modulation Index (*Tort et al., 2008*), generally increased during propofol anesthesia in Long-Evans Rats, generalized spike-and-wave seizures in GAERS rats, and psychedelic states in C57BL/6 mice, with no change during propofol anesthesia in human ET patients from both cortex to thalamus (*Figure 3—figure supplement 2*) and thalamus to cortex (*Figure 5—figure supplement 2*). These results suggest that low-to-high frequency cortical-thalamic information transfer is distinct from both conventional, non-spectral measures of directed information transfer, as well as from conventional measures of cross-frequency coupling, which only take into account linear and same-time interactions. As such, the strength of low-to-high frequency bidirectional cortical-thalamic information transfer is a specific and novel hallmark of conscious brain states.

## Cross-frequency information transfer between the cortex and thalamus is supported by edge-of-chaos criticality: mean-field modeling results

Based on our prior work indicating that the brain's information processing capacity during conscious states is supported by the proximity of slow cortical electrodynamics to edge-of-chaos criticality (*Toker et al., 2022*), we hypothesized that these changes in cross-frequency cortical-thalamic information transfer across brain states might be mediated by transitions of slow thalamocortical electrodynamics away from or closer to the edge-of-chaos critical point, or the phase transition from stable to chaotic dynamics. To test this hypothesis, we first developed a mean-field model of the electrodynamics of the

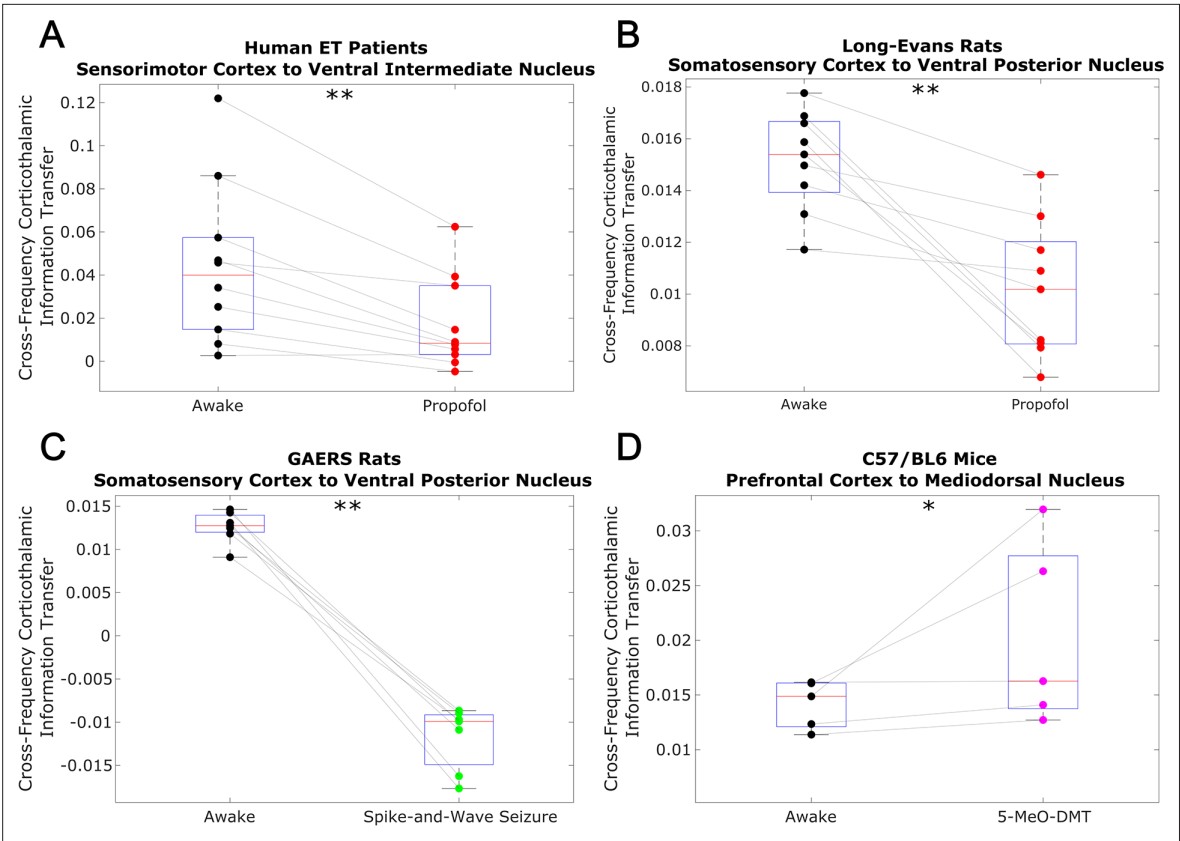

**Figure 3.** Low-to-high frequency information transfer from cortex to thalamus is diminished during unconsciousness and enhanced during psychedelic states. Using a spectrally resolved measure of directed information transfer (see Materials and methods), we found that the strength of information transferred from cortical $\delta/\theta/\alpha$ waves (~1–13 Hz) to thalamic high $\gamma$ waves (52–104 Hz) is significantly reduced during unconsciousness induced by propofol anesthesia (**A–B**) and generalized spike-and-wave seizures (**C**). Conversely, the strength of this low-to-high frequency corticothalamic information transfer is significantly increased during psychedelic states induced by 5-MeO-DMT (**D**). *p<0.05, **p<0.01, significance assessed using a one-tailed Wilcoxon signed-rank test.

The online version of this article includes the following figure supplement(s) for figure 3:

**Figure supplement 1.** Non-spectrally resolved transfer entropy from cortex to thalamus does not track consciousness.

**Figure supplement 2.** Low-to-high-frequency phase-amplitude coupling from cortex to thalamus does not track consciousness.

brain which could replicate these spectral patterns of cortical-thalamic information transfer observed in nearly all subjects/animals during waking states, and which could moreover replicate diverse, known features of neural electrodynamics. The reason we must first construct a mean-field model is because the presence and degree of chaos in any system can only be calculated with (some) certainty in a simulation, where noise and initial conditions can be precisely controlled in the estimation of the system's largest Lyapunov exponent (LLE) - a mathematically formal measure of chaoticity which quantifies how quickly initially similar system states diverge. It is for this reason that the study of chaos in biology should in general be paired with realistic simulations of the biological system of interest (***Glass and Mackey, 1988***; ***Toker et al., 2020***). Accordingly, we used Bayesian-genetic optimization to tune the parameters of a mean-field model of the basal ganglia-thalamo-cortical system (***Figure 6***) such that it generated biologically realistic large-scale neural electrodynamics across waking, anesthesia, and spike-and-wave seizure states (see Materials and methods and ***Figure 6—figure supplements 1–3*** for details).

The resulting simulations, using model parameters generated by our machine learning approach (see Materials and methods and ***Figure 6—figure supplements 1–3***) exhibited a broad range of biologically realistic features (***Figure 7***). First, our simulated cortical LFPs for the waking state exhibited spectral peaks at all canonical frequency bands, with the strongest peak in the $\alpha$ (8–13 Hz) range (***Figure 7—figure supplement 1***). Moreover, mean firing rates for each brain region in the model

closely matched known region-specific firing rates in mammals (**Table 2**). Furthermore, as in the real brain (**Ray et al., 2008**), there was a significant, positive correlation between fluctuations in our model's cortical firing rate and fluctuations in the amplitude of high-frequency (60–200 Hz) simulated cortical LFP activity (*r*=0.175, p=1.1e-35). Finally, recapitulating our novel empirical results (**Table 1**), our simulated cortical and thalamic LFPs exhibited significant, cross-frequency information transfer from thalamus to cortex (harmonic mean across 10 runs with different initial conditions $\mathring{p} = 0.0112$) and from cortex to thalamus ($\mathring{p} = 0.011$).

Beyond our simulation of the waking state, our anesthesia simulation likewise exhibited a broad range of biologically realistic features. First, in line with empirical results (**Figure 4**), at a 100% anesthetic 'dose', our simulated cortical LFPs exhibited increased low-frequency power and decreased high-frequency power relative to the simulated LFPs corresponding to the waking state (**Figure 7—figure supplement 2**). Moreover, increasing simulated 'doses' of simulated anesthesia effect recapitulated well-established dose-response trajectories of GABAergic anesthetics, including a continual decline in cortical firing rates (**Bastos et al., 2021**; **Figure 7—figure supplement 3A**) and LFP information-richness (**Frohlich et al., 2021**; **Figure 7—figure supplement 3B**), a continual rise in the power of low-frequency activity (**Billard et al., 1997**; **Figure 7—figure supplement 3**), and a transition to burst suppression followed by isoelectricity and cessation of firing at very high doses (**Ching and Brown, 2014**; **Figure 7**). Moreover, in line with both prior modeling (**Steyn-Ross et al., 2013**) and empirical (**Toker et al., 2022**) work, our simulated LFPs in the anesthesia state were more strongly chaotic than simulated cortical LFPs in the waking state (**Figure 7—figure supplement 3**). Furthermore, though these features were not explicitly selected for in our parameter optimization, our simulated anesthesia effect yielded several additional biologically realistic features, including the generation of LFPs with increasingly steep spectral slopes (**Colombo et al., 2019**; **Lendner et al., 2020**; **Figure 7—figure supplement 3E**), as well as prolonged inhibitory postsynaptic potentials (IPSPs) at excitatory cortical and thalamic relay cells relative to our waking simulation (**Kitamura et al., 2003**; **Hindriks and van Putten, 2012**; **Hutt and Longtin, 2010**; **Noroozbabaee et al., 2021**; **Figure 7—figure supplement 4**).

Finally, our generalized spike-and-wave seizure simulation likewise recapitulated several established biological features of seizures, including a large rise in cortical firing rates (**Figure 7—figure supplement 5A**; although cortical firing rates in our seizure simulation were considerably higher than in empirical data from GAERS rats **Jarre et al., 2017**) and a loss in the information-richness of cortical LFPs (**Mateos et al., 2018**; **Figure 7—figure supplement 5B**). In addition, following both prior empirical (**Toker et al., 2022**) and modeling (**Steyn-Ross et al., 2013**; **Breakspear et al., 2006**) results, our simulated LFPs in the seizure state were periodic, that is were insensitive to small perturbations (**Figure 7—figure supplement 5**). Example traces of cortical LFPs

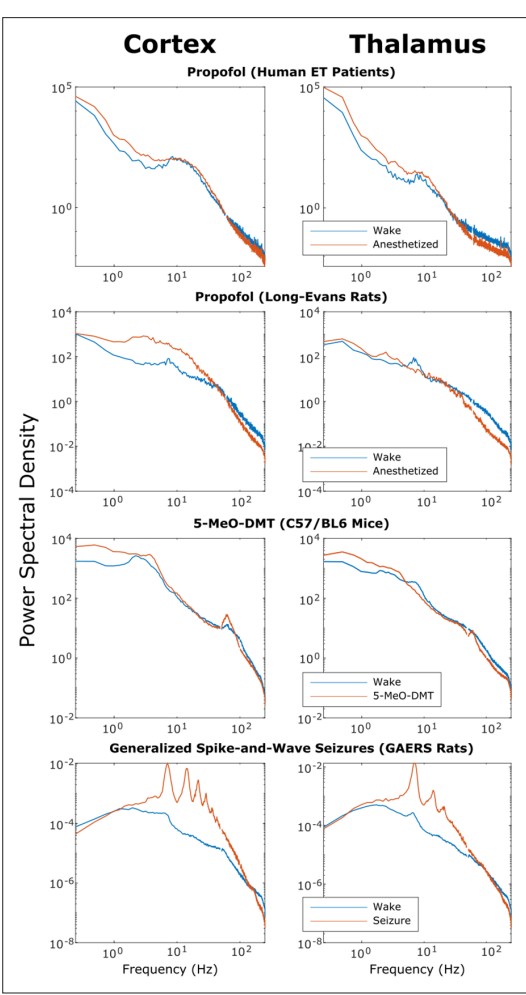

**Figure 4.** Power spectra of thalamic and cortical electrodynamics during waking, anesthesia, psychedelic, and seizure states. We here plot the cross-subject median power spectral densities (estimated using Welch's method) for all brain states. Note that both propofol and 5-MeO-DMT increased spectral power in the slow/delta range (≤4 Hz) and decreased spectral power above 80 Hz in both cortex and thalamus, despite opposing effects on cross-frequency corticothalamic information transfer (**Figure 3**).

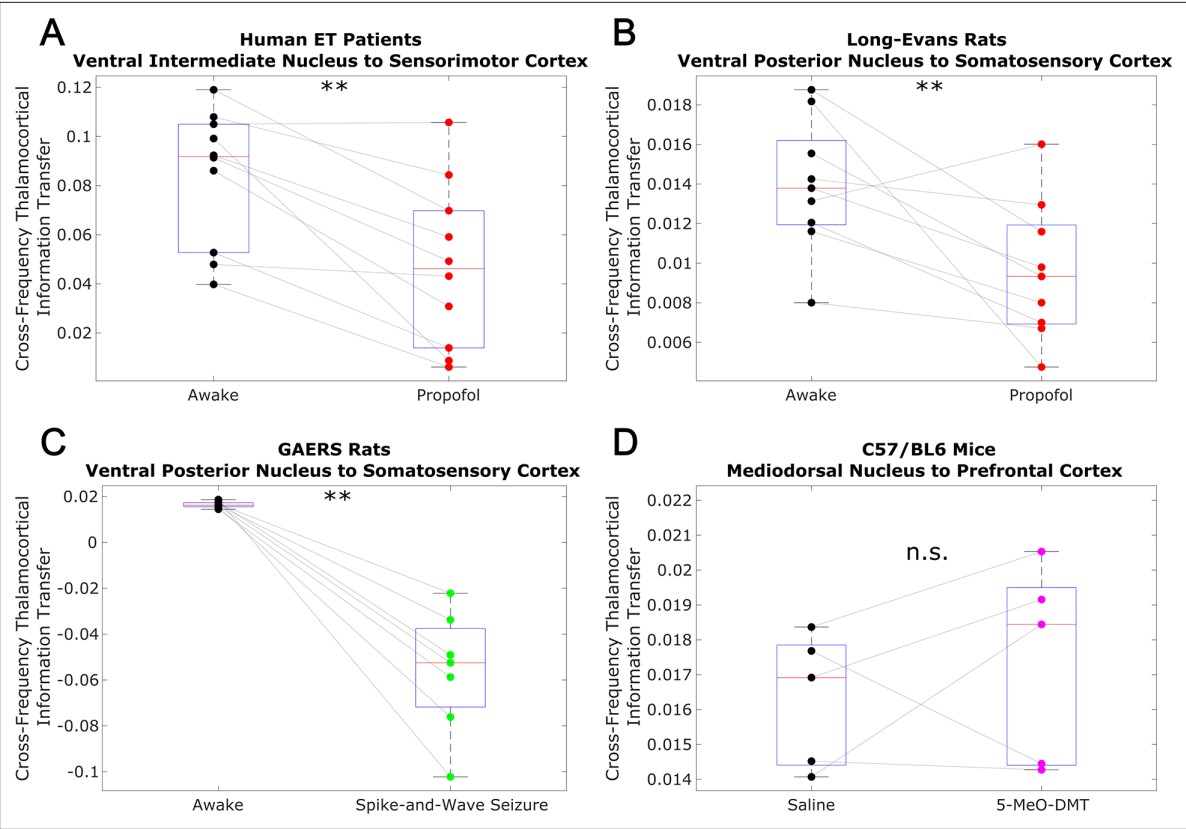

**Figure 5.** Low-to-high frequency information transfer from thalamus to cortex is diminished during unconsciousness. Similar to the results we observed for communication from the cortex to the thalamus (*Figure 3*), we found that strength of information transferred from thalamic $\delta/\theta/\alpha$ waves (~1–13 Hz) to cortical high $\gamma$ waves (52–104 Hz) is significantly reduced during unconsciousness induced by propofol anesthesia (**A–B**) and generalized spike-and-wave seizures (**C**). Unlike corticothalamic information transfer (*Figure 3*), however, the strength of this low-to-high frequency information transfer from the thalamus to cortex does not change significantly during psychedelic states induced by 5-MeO-DMT (**D**). *p<0.05, **p<0.01, significance assessed using a one-tailed Wilcoxon signed-rank test.

The online version of this article includes the following figure supplement(s) for figure 5:

**Figure supplement 1.** Non-spectrally resolved transfer entropy from thalamus to cortex does not track consciousness.

**Figure supplement 2.** Low-to-high-frequency phase-amplitude coupling from thalamus to cortex does not track consciousness.

from our simulations are plotted in *Figure 7*. Parameters for the three simulated brain states are listed in *Supplementary file 3*.

With these sufficiently realistic simulations of large-scale neural electrodynamics in hand, we used our mean-field model to assess, in silico, the relationship between edge-of-chaos criticality and bidirectional, cross-frequency information transfer between the cortex and thalamic relay nuclei. To do so, we generated LFPs at 50 increasing 'doses' of simulated anesthetic effect and 50 increasing strengths of seizure effect, relative to our normal waking simulation (see Materials and methods). The resulting parameter sweep yielded simulated cortical LFPs with a wide range of LLEs, including several near-critical LFPs (i.e. simulated LFPs with an estimated LLE near zero, indicating neither exponential divergence nor convergence of initially similar system states). Consistent with our predictions, we found that there was a clear peak of bidirectional, cross-frequency information transfer between our simulated cortical and thalamic LFPs when our simulated thalamocortical electrodynamics were poised near the edge-of-chaos critical point (*Figure 8A–B*). We found that bidirectional cross-frequency information transfer decayed as the (simulated) anesthetic effect was increased, which generated increasingly chaotic thalamocortical LFPs; likewise, cross-frequency information transfer between thalamus and cortex decayed as the (simulated) seizure effect was increased, which generated increasingly periodic LFPs, as shown in *Figure 8A–B*. Although these results offer compelling theoretical evidence for a relationship between edge-of-chaos criticality and the strength of cross-frequency information

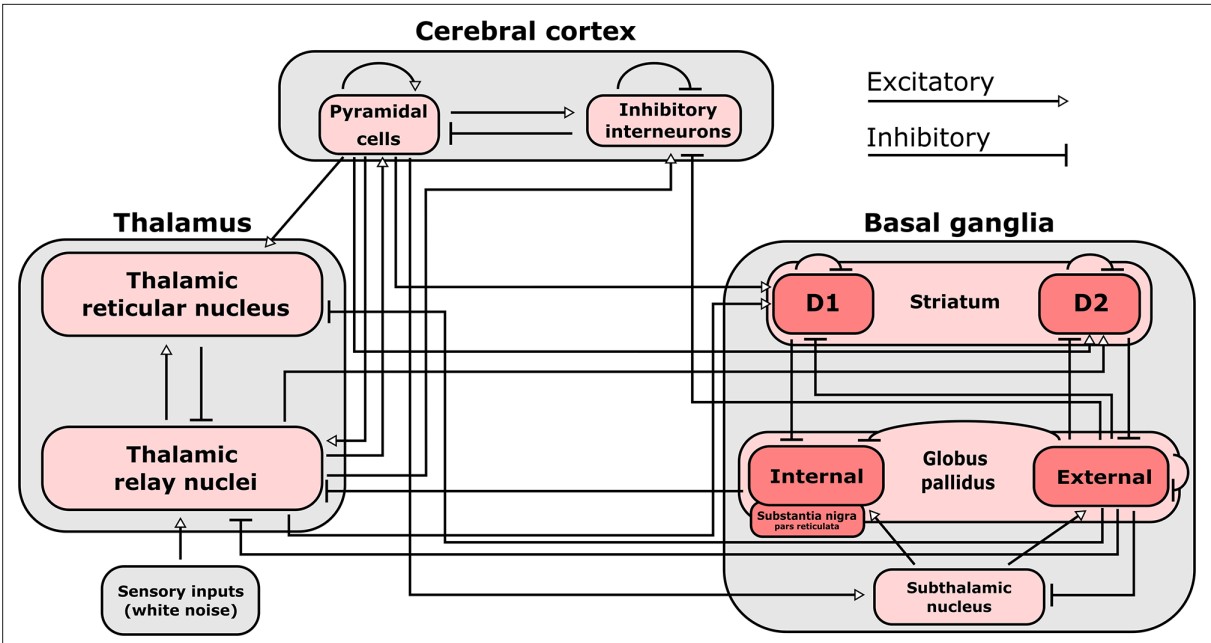

**Figure 6.** Connections included in our mean-field model of the macro-scale electrodynamics of the basal ganglia-thalamo-cortical system. We here plot the structural connectivity in out mean-field model. Note that the internal globus pallidus and the substantia nigra pars reticulata, which are both inhibitory output nuclei of the basal ganglia, are treated as a single structure. See *Table 2* for the mean firing rates for each neural population in the model, alongside known region-specific firing rates in multiple mammalian species. See *Supplementary file 3* for parameters describing the properties of each neural population, as well as parameters describing the propagation of electric fields along each anatomical connection.

The online version of this article includes the following figure supplement(s) for figure 6:

**Figure supplement 1.** Method for optimizing parameters of the waking simulation.

**Figure supplement 2.** Method for optimizing parameters of the anesthesia simulation.

**Figure supplement 3.** Method for optimizing parameters of the seizure simulation.

transfer between the thalamus and cortex, LLEs cannot be accurately estimated in empirical data, and therefore alternative chaos detection algorithms are required in order to empirically test this relationship between chaoticity and cross-frequency cortical-thalamic communication in real brains. Because the K-statistic of the modified 0–1 chaos test has previously been demonstrated to accurately estimate chaoticity from empirical time-series recordings (*Toker et al., 2020*), we tested whether the K-statistic could accurately track chaoticity in our mean-field simulation. Indeed, when applied to simulated thalamocortical LFPs bandpass filtered between 1–13 Hz (matching the slow frequencies of cortical-thalamic information transfer identified here), the K-statistic was significantly correlated with the estimated largest Lyapunov exponent of our simulated LFPs ($\rho$ =0.76, p=0), and could moreover recapitulate the observed relationship between thalamocortical chaoticity and cross-frequency cortical-thalamic information transfer in our mean-field model, as shown in *Figure 8C–D*. This indicates that the K-statistic of the modified 0–1 chaos test can be used to test the predicted relationship between proximity to edge-of-chaos criticality and the strength of cross-frequency cortical-thalamic information transfer in real brain data.

## Cross-frequency information transfer between the cortex and thalamus is supported by edge-of-chaos criticality: empirical results

Because the K-statistic of the 0–1 chaos test can be calculated from empirical neural data, we applied the test to our electrophysiology recordings, bandpass filtered between 1 and 13 Hz. Confirming our predictions, the empirical results recapitulated the relationship between thalamocortical chaoticity and cortical-thalamic cross-frequency information transfer observed in our mean-field model (*Figure 8*), with maximal information transfer occurring for intermediary levels of estimated chaoticity (presumably reflecting proximity to edge-of-chaos criticality; *Figure 9*). Importantly, replicating both prior

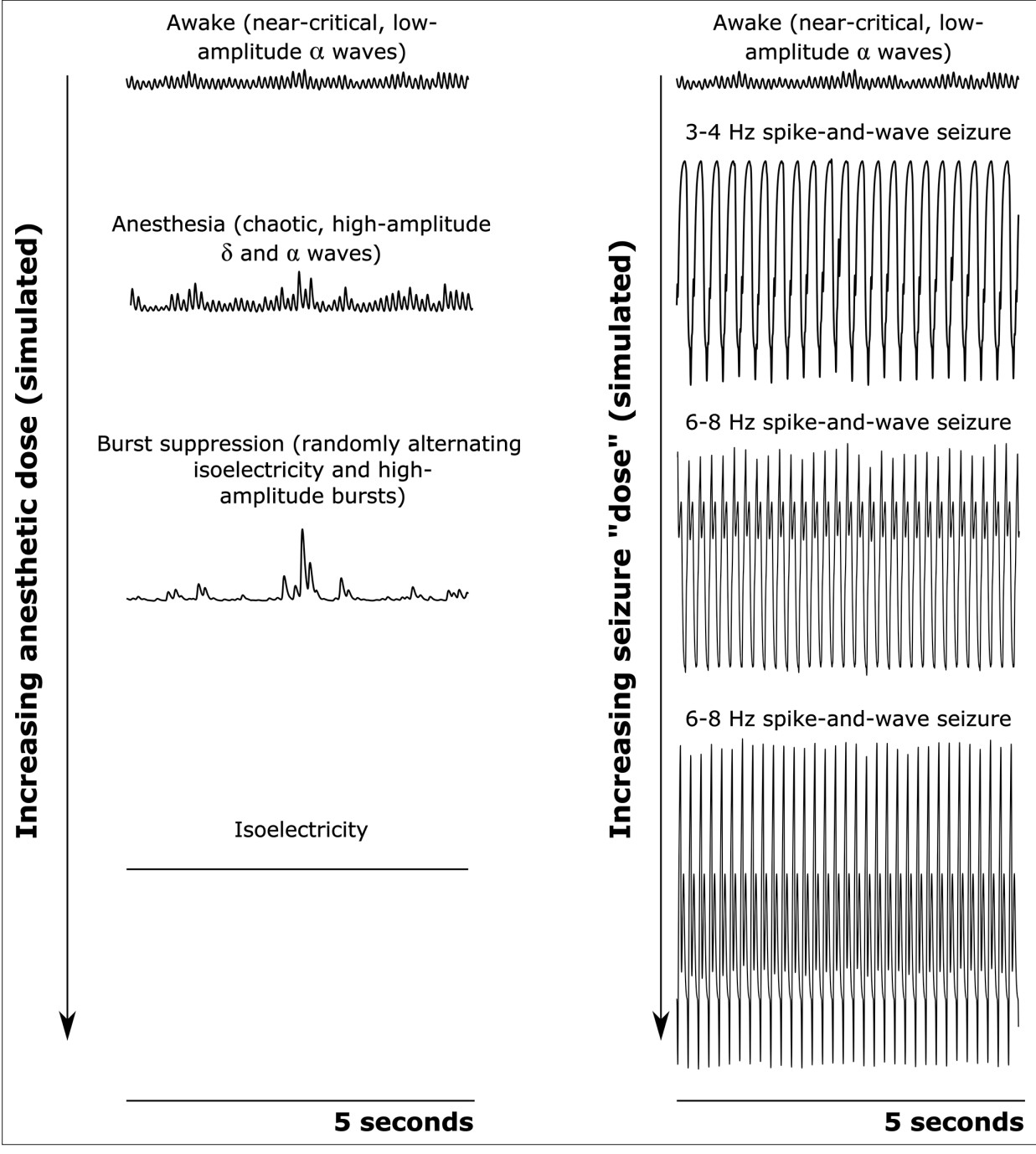

**Figure 7.** Simulated cortical local field potentials as a function of increasing anesthetic or seizure 'dose'. We here plot example time-traces of our simulated cortical local field potentials (LFPs). Note that all data plotted here are on the same scale. For our awake simulation (top), the mean-field model generates near-critical, weakly chaotic, low-amplitude oscillations dominated by $\alpha$ waves (8–13 Hz), with significant bidirectional cross-frequency information transfer between the cortex and thalamus (as observed in our empirical data). With increasing anesthetic dose (left), the simulated cortical LFP transitions to chaotic, high-amplitude $\delta$ waves (1–4 Hz) and $\alpha$ waves. At a higher dose, the simulated cortical LFP transitions to burst suppression-like dynamics, which are characterized by stochastic switching between isoelectricity and high-amplitude bursts. Finally, at the highest anesthetic doses, the simulated cortical LFP transitions to isoelectricity. This simulated anesthetic dose-response trajectory closely mirrors well-established empirical dose-response trajectories. For our seizure simulation (right), increasing 'doses' first push the cortical LFP into a 3–4 Hz spike-and-wave seizure (which is characteristic of human epilepsy patients), followed by a 6–8 Hz spike-and-wave seizure (which is characteristic of rodent models of epilepsy, including the GAERS rats studied here).

The online version of this article includes the following figure supplement(s) for figure 7:

*Figure 7 continued*

**Figure supplement 1.** Power spectrum of the simulated, waking-state cortical local field potential.

**Figure supplement 2.** Spectral changes in the anesthesia simulation.

**Figure supplement 3.** Dose-response effects of simulated anesthesia.

**Figure supplement 4.** Inhibitory postsynaptic potentials in the anesthesia simulation.

**Figure supplement 5.** Dose-response effects of simulated seizures.

simulation and empirical results (*Toker et al., 2022*) as well as the novel simulation results presented above (*Figure 8*, *Figure 7—figure supplement 3*), we found that GABAergic anesthesia destabilized slow thalamocortical electrodynamics in both humans (p=0.065, one-tailed Wilcoxon signed-rank test comparing patients' cross-trial median K-statistic during waking states versus propofol anesthesia states) and rats (p=0.002, one-tailed Wilcoxon signed-rank test). Conversely, slow thalamocortical activity became periodic or hyper-stable during generalized spike-and-wave seizures (p=0.0078, one-tailed Wilcoxon signed-rank test). Finally, 5-MeO-DMT moderately stabilized cortical electrodynamics (p=0.0312, one-tailed Wilcoxon signed-rank test), which is consistent with prior results showing that psychedelics tune slow neural electrodynamics closer to edge-of-chaos criticality, and do so by

**Table 2.** Mean firing rates of each brain region in the waking state of the mean-field model (in spikes/s), compared to empirical ranges of firing rates from multiple mammalian species.

Note that the GPi and SNr were treated as a single population, and thus have the same firing rate. Mean firing rates for each simulated brain region were within or near known physiological ranges, and were tuned to be as such using Bayesian-genetic optimization. GPi = internal globus pallidus, SNr = substantia nigra pars reticulata, GPe = external globus pallidus, STN = subthalamic nucleus, TRN = thalamic reticular nucleus.

| | Model | Monkeys | Rats | Mice | Humans | Cats |
|---|---|---|---|---|---|---|
| Cortex | 7.7 | 5-20* | 2-5[†] | 2-11[‡] | 1-10[§] | |
| Striatum | 5.19 | 4-7[¶] | 1-7** | | | |
| GPi | 75.72 | 60-90[††] | 15-20[‡‡] | | | |
| SNr | 75.72 | 50-70[§§] | | | | |
| GPe | 30.85 | 16-70[¶¶] | 16-115*** | 1-64[†††] | | |
| STN | 13.62 | 20-30[‡‡‡] | 8-11[§§§] | | | |
| Relay nuclei | 16.39 | 5-25[¶¶¶] | | | 10-20**** | |
| TRN | 15.41 | | | 4-64[††††] | | 20-30[‡‡‡‡] |

[*]*Goldberg et al., 2002*; *Wannier et al., 1991*.
[†]*Dejean et al., 2008*.
[‡]*Fan et al., 2016*.
[§]*Paulk et al., 2022*.
[¶]*Goldberg et al., 2002*; *Yoshida, 1991*.
[**]*Dejean et al., 2008*; *Kiyatkin and Rebec, 1996*.
[††]*DeLong, 1971*; *Georgopoulos et al., 1983*; *Heimer et al., 2002*; *Kimura et al., 1996*.
[‡‡]*Dejean et al., 2008*.
[§§]*DeLong et al., 1983*; *Schultz, 1986*.
[¶¶]*Bugaysen et al., 2010*; *DeLong, 1971*; *DeLong et al., 1985*; *Georgopoulos et al., 1983*; *Heimer et al., 2002*.
[***]*Deister et al., 2013*; *Pan and Walters, 1988*.
[†††]*Akopian et al., 2016*.
[‡‡‡]*Benazzouz et al., 2002*; *Georgopoulos et al., 1983*.
[§§§]*Kreiss et al., 1997*.
[¶¶¶]*Ramcharan et al., 2005*.
[****]*Molnar et al., 2005*.
[††††]*Campbell et al., 2020*; *Lewis et al., 2015*; *Mukhametov et al., 1970*; *Wimmer et al., 2015*.
[‡‡‡‡]*Steriade et al., 1986*.

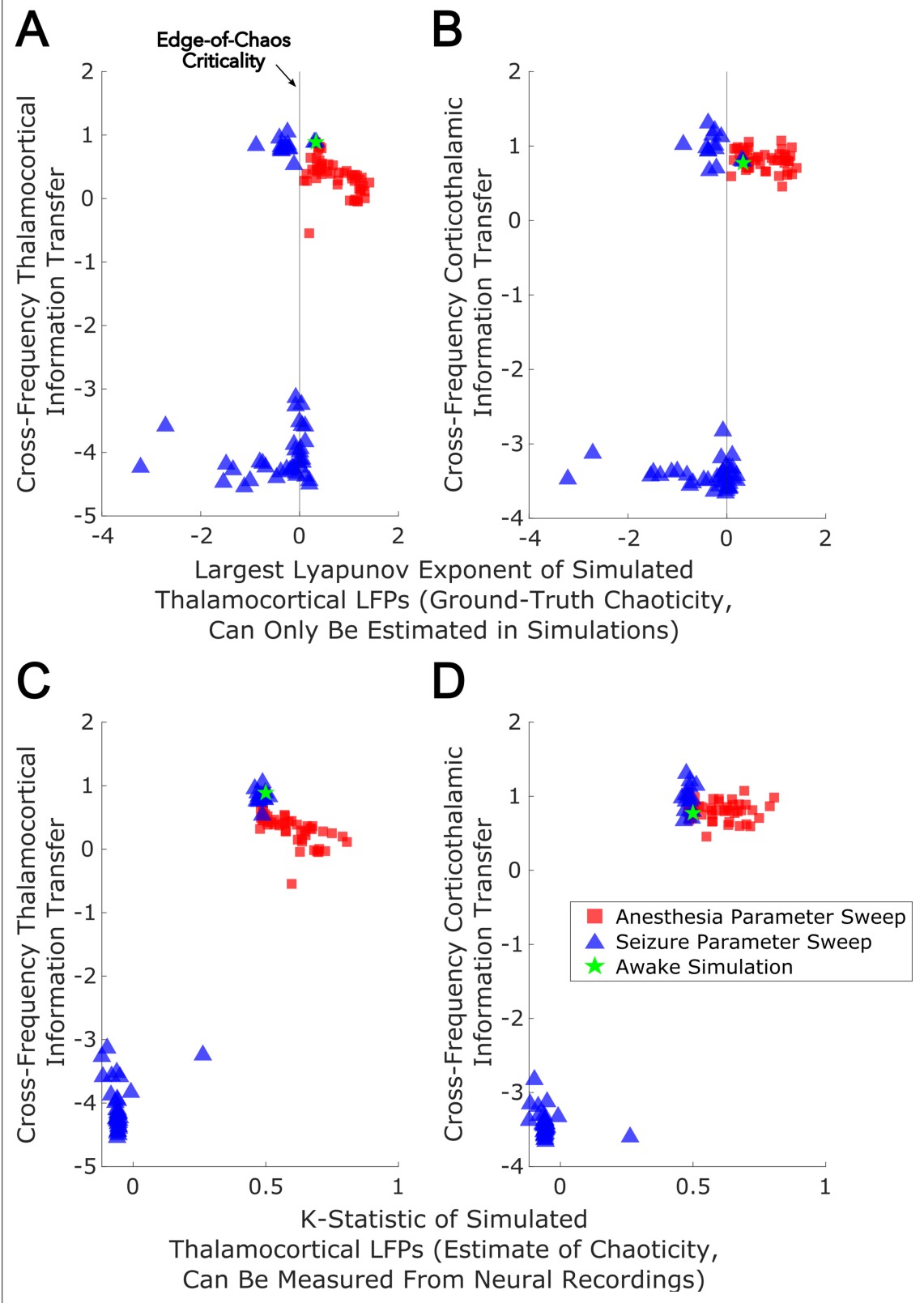

**Figure 8.** Edge-of-chaos criticality supports cross-frequency thalamic-cortical information transfer in a mean-field model. We performed parameter sweeps for different 'doses' of simulated anesthetic (red square) and seizure (blue triangle) effects. For each 'dose', we calculated the median estimated largest Lyapunov exponent (LLE) of simulated thalamocortical LFPs across 10 runs, and plotted the median strength of cross-frequency thalamocortical (**A**) and corticothalamic (**B**) information transfer as a function of those median LLEs. We found a clear peak in the strength of bidirectional cross-

*Figure 8 continued on next page*

*Figure 8 continued*

frequency cortical-thalamic information transfer when our simulated thalamocortical electrodynamics were poised near edge-of-chaos criticality (the vertical lines at LLE = 0). We further found that the strength of this bidirectional, cross-frequency information transfer decayed in both the periodic phase (negative LLEs) with increasing seizure effect and the chaotic phase (positive LLEs) with increasing anesthetic effect. However, because this decay was exponentially faster in the periodic phase, we here plotted the bi-symmetric log-transform (*Webber, 2013*) of our results for the sake of visualization. Because LLEs can only be estimated with some accuracy in simulations, we also calculated the estimated the median chaoticity of the low-frequency (1–13 Hz) component of our simulated cortical and thalamic LFPs using the K-statistic of the modified 0–1 chaos (which can be measured from real neural recordings). We plotted those results against the (bi-symmetric log-transformed) median strength of cross-frequency thalamocortical (**C**) and corticothalamic (**D**) information transfer, and observed the same overall relationship between chaoticity and bidirectional cross-frequency information transfer, suggesting that this relationship can be evaluated in real neural recordings.

approaching the critical point from the chaotic side of the edge (*Toker et al., 2022*). Finally, while the estimated chaoticity of low-frequency (1–13 Hz) thalamocortical electrodynamics varied significantly across brain states (p=0.0001) as assesed by permutation-based nonparametric ANCOVA (*Scheiner and Gurevitch, 2001*), this variance could not be explained by changes to spectral power in this frequency range in the thalamocortical system across brain states (*Supplementary file 4*).

## Discussion

We here identified a highly preserved spectral pattern of cross-frequency information transfer between the cortex and thalamus across species during waking states, wherein information sent from one brain structure at low frequencies (~1–13 Hz) is encoded by the other at high frequencies (52–104 Hz). We moreover showed that this pattern of information transfer is disrupted during unconscious states, possibly because low-frequency thalamocortical electrodynamics diverge from edge-of-chaos criticality during these states. Conversely, we showed that this low-to-high frequency information transfer from the cortex to the thalamus is enhanced during psychedelic states, possibly because slow

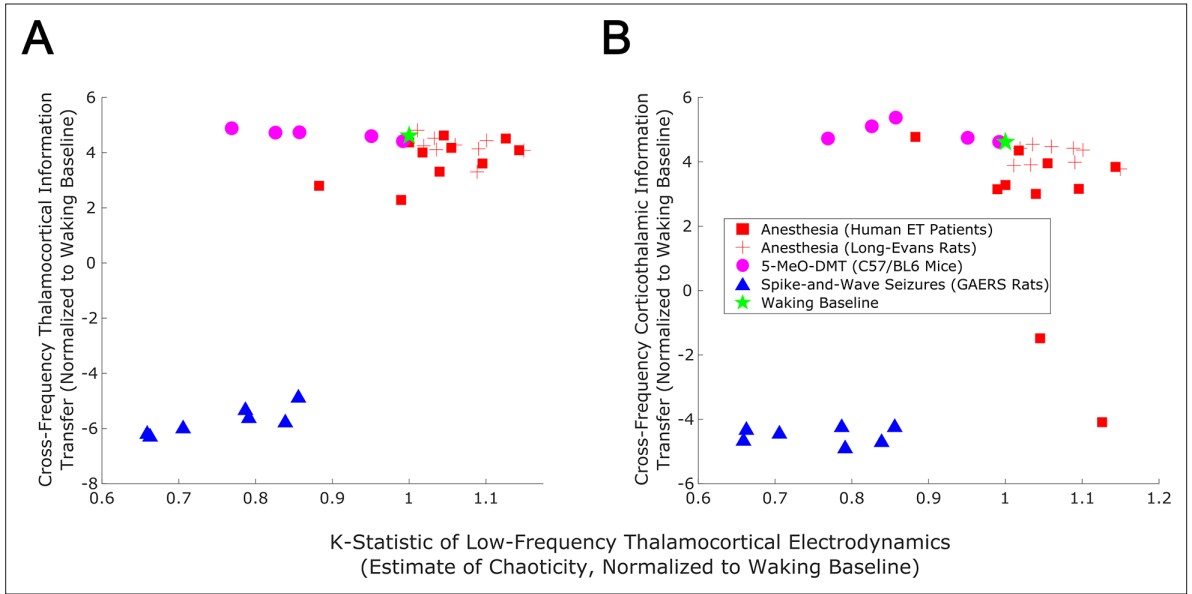

**Figure 9.** Empirical evidence that edge-of-chaos criticality supports cross-frequency thalamic-cortical information transfer during conscious states. We here plot the median strength of cross-frequency thalamocortical (**A**) and corticothalamic (**B**) information transfer across brain states (normalized to each patient's or animal's waking baseline, and bi-symmetrically log-transformed) as a function of the median estimated chaoticity of the low-frequency (1–13 Hz) component of thalamic and cortical electric field recordings (also normalized to waking baselines). We found the same trend as in our mean-field model (*Figure 8*), with bidirectional cross-frequency information transfer exhibiting the most pronounced decay as thalamocortical electrodynamics hyper-stabilize in the generalized spike-and-wave seizure state. The strength of bidirectional cross-frequency information transfer also decays, though not as quickly, as thalamocortical electrodynamics become increasingly chaotic in the GABAergic anesthesia state. Conversely, the strength of cross-frequency information transfer from the cortex to the thalamus, but not from the thalamus to the cortex, increases as thalamocortical electrodynamics moderately stabilize in the 5-MeO-DMT psychedelic state, presumably reflecting a transition closer to edge-of-chaos criticality relative to normal waking states, which are near-critical but weakly chaotic.

thalamocortical electrodynamics are tuned closer to edge-of-chaos criticality during these states (and approach this critical point from the chaotic side of the edge, where our evidence suggests normal waking slow thalamocortical electrodynamics lie). Note that we did not observe a significant increase in cross-frequency information transfer from the thalamus to cortex during psychedelic states, though this may be due to our small sample size of animals in this condition (n=5).

To provide theoretical evidence for this relationship between edge-of-chaos criticality and cross-frequency cortical-thalamic information transfer, we used Bayesian-genetic optimization to tune a mean-field model of the electrodynamics of the full basal ganglia-thalamo-cortical system, so that it could recapitulate diverse aspects of real neural electrodynamics while using biologically realistic parameters (see Materials and methods). In the context of simulating the waking brain, our model is the first to encapsulate cross-frequency information transfer between the thalamus and cortex, a phenomenon our empirical data and analyses have uniquely identified. Moreover, prior models of the full basal ganglia-thalamo-cortical system during healthy, waking states have largely relied on the assumption that neural dynamics in these states are typified by a stable fixed point (possibly) perturbed by noise (*van Albada and Robinson, 2009*; *Holgado et al., 2010*; *Pavlides et al., 2012*) - an assumption not supported by our current (*Supplementary file 5*) or prior (*Toker et al., 2022*) findings, and our model, which generates nonlinear oscillatory behavior, accords with these findings. Our model of the full basal ganglia-thalamo-cortical system is also the first, to our knowledge, to accurately recapitulate waking state firing rates in the cortex and all major subcortical areas (*Table 2*), while also generating oscillatory dynamics in the regime of weak chaos, near edge-of-chaos criticality. Finally, our mean-field model of the basal ganglia-thalamo-cortical system is also unique in its inclusion of pallido-thalamic, pallido-cortical, and pallido-striatal projections (see Materials and methods). Additionally, while prior models have associated the emergence of alpha oscillations with Hopf bifurcations (*Grimbert and Faugeras, 2006*; *Spiegler et al., 2010*), our model is the first (to our knowledge) to do so specifically in the proximity of edge-of-chaos criticality. In the context of anesthesia simulations, our model is unique in its inclusion of the basal ganglia, as prior mean-field models of neural electrodynamics during the anesthetized state have largely focused on either just cortical (*Steyn-Ross et al., 2013*; *Kuhlmann et al., 2016*; *Bojak and Liley, 2005*; *Liley and Walsh, 2013*; *Molaee-Ardekani et al., 2007*) or just thalamo-cortical dynamics (*Ching and Brown, 2014*; *Hindriks and van Putten, 2012*; *Noroozbabaee et al., 2021*). Similarly, while strong chaos has been identified in cortical-only models of anesthesia (*Steyn-Ross et al., 2013*), our model seeks to offer a broader perspective by introducing this phenomenon in a context that incorporates several subcortical structures. Given these unique features of our model as well as its broad biological realism, we believe that our model of the basal ganglia-thalamo-cortical system - or perhaps future versions of it, which are even more closely matched to empirical results from multiple brain states - may be a fruitful tool for future in silico studies of possible interventions to modulate consciousness.

Although both our empirical and simulated thalamocortical electrodynamics show clear evidence of cross-frequency cortical-thalamic information transfer, and that the strength of this cross-frequency information transfer is supported by the proximity of thalamocortical electrodynamics to edge-of-chaos criticality, much work remains to be done to explain this frequency-specific communication pattern during conscious states. In other words, the precise code of cross-frequency communication remains to be determined. It is possible, for example, that this code will be related to mechanisms that are by now well-established in the neuroscience literature, such as the modulation of the amplitude of high-frequency activity by the phase of low-frequency activity (*Canolty and Knight, 2010*). Indeed, our observation of cross-frequency information transfer between thalamus and cortex is, at least conceptually, consistent with prior evidence of low-to-high frequency phase-amplitude coupling between these regions during waking states (*Fitzgerald et al., 2013*; *Malekmohammadi et al., 2019*; *Opri et al., 2019*; *Malekmohammadi et al., 2015*); however, it is important to note that, unlike the strength of directed cross-frequency information transfer, the strength of phase-amplitude coupling did not consistently vary as a function of brain state (*Figure 3—figure supplement 2*, *Figure 5—figure supplement 2*), which suggests that these are somewhat distinct phenomena. It may also be that cross-frequency cortical-thalamic information transfer could rely on coding mechanisms which have not yet been explored in the neuroscience literature, but which have been explored in the communications engineering literature, such as low-to-high-frequency information transfer using the harmonic backscattering of low-frequency signals (*An et al., 2018*).

We note several limitations to the work done here, and fruitful areas for further investigation. First, although our evidence suggests that cortical-thalamic information transfer from ~1–13 Hz to 52–104 Hz is a hallmark of conscious brain states, it is currently unclear whether information transmission along other frequency bands likewise signposts consciousness. Indeed, in our preliminary exploratory sweep of patterns of information flow (*Figure 2*), we found that although thalamus-to-cortex communication may be uniquely characterized by this spectral pattern, communication from the cortex to the thalamus may be more broadband. Thus, in future work, it will be important to more completely characterize the strength and statistical significance of cortical-thalamic information transfer along other frequency bands in both conscious and unconscious brain states. We also stress that currently, varying degrees of chaoticity - and therefore proximity to edge-of-chaos criticality - can only be detected with some certainty in simulations. The modified 0–1 chaos test, which we used here as an empirical test of chaoticity, is a relatively robust method for chaos detection (*Toker et al., 2020*), correlates well with ground-truth chaoticity in our mean-field model, and reproduces the relationship between chaoticity and cross-frequency cortical-thalamic information transfer observed in our simulations; but, the test's results may be affected by features of a signal, such as noise, which are unrelated to ground-truth chaoticity. For this reason, it will be imperative to develop additional methods for assessing the chaoticity of thalamocortical electrodynamics in order to confirm or falsify the observations reported here. It will moreover be important to study how generalized seizures, anesthesia, and psychedelics affect information transfer between the cortex and other subcortical regions which have been implicated in the loss and recovery of consciousness, such as the basal ganglia (*Miyamoto et al., 2019*; *Deransart et al., 2000*; *Chen et al., 2015b*; *DiCesare et al., 2020*; *Crone et al., 2017*; *Lutkenhoff et al., 2015*; *Lutkenhoff et al., 2020*; *Lazarus et al., 2012*; *Qiu et al., 2016a*; *Vetrivelan et al., 2010*; *Qiu et al., 2016b*, *Qiu et al., 2010*), and how that in turn relates to the proximity of neural electrodynamics to edge-of-chaos criticality. In a similar vein, it will also be important to test whether the observed phenomena extend to other states of unconsciousness (e.g. coma and vegetative states) and other psychedelic states (e.g. induced by lysergic acid diethylamide or psilocybin).

## Materials and methods
### Mean-field model of the electrodynamics of the basal ganglia-thalamocortical system

To study the relationship between edge-of-chaos criticality and cross-frequency cortical-thalamic information transfer, and how that might change during GABAergic anesthesia and generalized spike-and-wave seizures, we developed a modified version of the mean-field model of the basal ganglia-thalamocortical system described by *van Albada and Robinson, 2009*. Although our empirical analysis focuses on thalamo-cortical interactions, we chose a model which includes the basal ganglia because of recent evidence that the basal ganglia (perhaps via their influence on the thalamus and cortex) are involved in the loss and recovery of consciousness from generalized seizures (*Miyamoto et al., 2019*; *Deransart et al., 2000*; *Chen et al., 2015b*), anesthesia (*DiCesare et al., 2020*; *Crone et al., 2017*), vegetative and minimally conscious states (*Lutkenhoff et al., 2015*; *Lutkenhoff et al., 2020*), and sleep (*Lazarus et al., 2012*; *Qiu et al., 2016a*; *Vetrivelan et al., 2010*; *Qiu et al., 2016b*, *Qiu et al., 2010*).

The model simulates the average firing rate of several populations of neurons, which is estimated as the proportion of neurons within a population whose membrane potential is greater than their reversal potential, multiplied by the maximum possible firing rate for that population. Specifically, the average population activity $Q_a$ at location $r$ and time $t$ is modeled as a sigmoidal function of the number of cells whose potential $V_a$ is above the mean threshold potential $\theta$ of that population:

$$Q_a(r,t) = \frac{Q_a^{\max}}{1 + \exp[-(V_a(t) - \theta_a)/\sigma']}$$

(1)

where $Q_a^{max}$ is the maximum possible firing rate of that population and $\sigma'$ is the standard deviation of cell body potentials relative to the threshold potential. The change in mean cell potential $V_a$ is modeled as:

$$D_{\alpha\beta}(t)V_a(t) = \sum_b v_{ab}\phi_b(t - \tau_{ab})$$

(2)

where $v_{ab}$ is the number of synapses between the axons of population $b$ and dendrites of population $a$ multiplied by the typical change in the membrane potential of a cell in $a$ with each incoming electric pulse from $b$. $\Phi_b(t - \tau_{ab})$ is the rate of incoming pulses from $b$ to $a$, $\tau_{ab}$ is the time delay for signals traveling across axons from $b$ to $a$, and $D_{\alpha\beta}$ is the differential operator

$$D_{\alpha\beta}(t) = \frac{1}{\alpha\beta}\frac{d^2}{dt^2} + \left(\frac{1}{\alpha} + \frac{1}{\beta}\right)\frac{d}{dt} + 1 \tag{3}$$

where $\alpha$ is the decay rate of the cell membrane potential and $\beta$ is the rise rate of the neural membrane potential. In the original Robinson mean-field model, not only the duration, but also the peak $\eta$ of synaptic responses is scaled by $\alpha$ and $\beta$:

$$\eta(\alpha, \beta) = \frac{\alpha\beta}{\beta - \alpha}\left[\exp\left(-\alpha\frac{\ln(\beta/\alpha)}{\beta - \alpha}\right) - \exp\left(-\beta\frac{\ln(\beta/\alpha)}{\beta - \alpha}\right)\right] \tag{4}$$

However, since we are interested in modeling GABAergic anesthesia, which prolongs the duration of postsynaptic inhibition - an effect that can be simulated by modulating the synaptic decay rate $\alpha$ (*Hindriks and van Putten, 2012*; *Hutt and Longtin, 2010*) or potentially the rise rate $\beta$ - without altering the maximal postsynaptic chloride current (*Kitamura et al., 2003*), we followed prior modeling studies of anesthesia (*Hindriks and van Putten, 2012*; *Hutt and Longtin, 2010*; *Bojak and Liley, 2005*; *Noroozbabaee et al., 2021*) and modified the synaptic response $h$, such that its duration but not its peak is modulated by $\alpha$ and $\beta$:

$$h(t) = \frac{H}{\eta(\alpha, \beta)}\bar{h}(t) \tag{5}$$

where $\bar{h}(t)$ is the original synaptic response, and, following *Hindriks and van Putten, 2012*, $H$=31.5 $s^{-1}$. Finally, the outgoing mean electric field $\phi_{ab}$ from population $b$ to population $a$ is modeled with the widely used damped wave equation

$$\mathcal{D}_{ab}\phi_{ab}(r, t) = Q_b(r, t) \tag{6}$$

with

$$\mathcal{D}_{ab} = \left[\frac{1}{\gamma_{ab}^2}\frac{\partial^2}{\partial t^2} + \frac{2}{\gamma_{ab}}\frac{\partial}{\partial t} + 1 - r_{ab}^2\nabla^2\right] \tag{7}$$

where $r_{ab}$ is the spatial axonal range, $\gamma_{ab}$ is the temporal damping coefficient and equals $v_{ab}/r_{ab}$, and $\nabla^2$ is the Laplacian operator.

Importantly, apart from circuit connectivity described in the original van Albada and Robinson model, we included several additional known afferent projections from the globus pallidus externa (GPe; *Figure 6*), given the recent evidence for the importance of the GPe in particular in regulating the loss and recovery of consciousness (*Lazarus et al., 2012*; *Qiu et al., 2016a*; *Vetrivelan et al., 2010*; *Qiu et al., 2016b, Qiu et al., 2010*; *Zheng and Monti, 2019*). Specifically, in light of recent tracing studies in mice showing direct GABAergic projections from GPe to GABAergic cortical interneurons (*Saunders et al., 2015*; *Chen et al., 2015a*), as well as recent high angular resolution diffusion imaging showing direct projections from GPe to cortex in humans (*Zheng and Monti, 2019*), we added inhibitory connections from GPe to inhibitory cortical neurons. We also added direct inhibitory projections from GPe to thalamic relay nuclei, following recent human high angular resolution diffusion imaging results (*Zheng and Monti, 2019*). Moreover, following results from tracing studies in squirrel monkeys (*Hazrati and Parent, 1991*), we additionally added direct inhibitory projections from GPe to the thalamic reticular nucleus. We furthermore added inhibitory connections from GPe to both D1 and D2 striatal populations, based on extensive prior tracing studies showing pallidostriatal projections in rats (*Kuo and Chang, 1992*; *Staines et al., 1981*; *Kuo and Chang, 1992*; *Staines and Fibiger, 1984*; *Rajakumar et al., 1994*), cats (*Beckstead, 1983*), and monkeys (*Beckstead, 1983*; *Kita et al., 1999*; *Sato et al., 2000*).

The model thus constructed contains 185 free parameters. In the original model, van Albada and Robinson identified a parameter configuration within physiologically realistic bounds that produced stable fixed points of neuronal firing rates for each brain region, which can be analytically identified

using well-known mathematical tools. Under this approach, fluctuations of neuronal firing rates are generated via noise perturbations away from and back toward these stable fixed points. However, this approach assumes that macroscale neural electrodynamics are perfectly stable unless perturbed, which is contradicted by some empirical evidence: low-frequency electrodynamic oscillations have been observed in the absence of any sensory inputs or perturbations in isolated, deafferented cortex (*Timofeev et al., 2000*; *Lemieux et al., 2014*) and in deafferented thalamic reticular nucleus (*Steriade et al., 1987*), as well as in unperturbed cerebral organoids (*Trujillo et al., 2019*; *Samarasinghe et al., 2019*). Moreover, this modeling approach assumes that neural electrodynamic oscillations are predominantly stochastic, which our current (*Supplementary file 5*) and past (*Toker et al., 2022*) work suggests is not the case. In line with this broad empirical evidence for intrinsic low-frequency, nonlinear oscillatory electrical activity in the brain, other mean-field modeling approaches have sought instead to understand slow neural electrodynamics (in both waking and non-waking states) in terms of (often chaotic) nonlinear oscillations, rather than in terms of noise perturbations of stable fixed points (*Dafilis et al., 2001*; *Steyn-Ross et al., 2013*; *Freeman, 1987*). In accordance with this approach, we sought a physiologically realistic parameter configuration for waking brain states that would yield low-amplitude, oscillatory, weakly chaotic oscillations of local field potentials (LFPs), where the LFPs of a given neural population were simulated by taking the superposition of synaptic currents (*Buzsáki et al., 2012*), estimated as the sum of the absolute value of dendritic potentials of that population (*Mazzoni et al., 2015*). In addition to meeting this criterion of generating low-amplitude, weakly chaotic LFPs, we sought a parameter configuration for waking states which yields mean firing rates for all brain regions that match empirical data, which generates fluctuations in cortical firing rates that are correlated with fluctuations in the amplitude of high gamma (60–200 Hz) cortical LFP oscillations, and which additionally recapitulates the spectral patterns of bidirectional cortico-thalamic information transfer we identified in our empirical data. Because there are no methods for deriving such a parameter configuration analytically, and because the parameter space of the model is infinite (though bounded) and thus impossible to explore through a systematic parameter sweep, we used a Bayesian-genetic machine learning algorithm (*Lan et al., 2022*) to tune all parameters in the model to produce the desired dynamics (see Supplementary Methods and *Figure 6—figure supplements 1–3* for flowcharts describing the details of the Bayesian-genetic optimization).

Once we identified a parameter configuration for waking brain states (*Supplementary file 3*), we used that parameter configuration as the starting point for a search, using genetic optimization, for parameter configurations that would produce GABAergic anesthesia and generalized spike-and-wave seizure dynamics. For the seizure dynamics, we simply tuned the model's parameters to generate 2–8 Hz oscillations that are periodic and information-poor (as indexed by Lempel-Ziv complexity), which resulted in spike-and-wave behavior. For the anesthesia dynamics, we tuned the model's parameters to minimize the cortical firing rate while simultaneously generating information-poor, strongly chaotic LFPs that are dominated by large-amplitude slow/delta (<4 Hz) oscillations with low spectral power above 60 Hz. Once we identified a set of parameters for our awake simulation, our anesthesia simulation, and our spike-and-wave seizure simulation (*Supplementary file 3*), we used the following equation to produce a given parameter set $P$ at a particular 'dose' $D$ of simulated anesthetic or seizure effect:

$$P = P_0 (\frac{P_1}{P_0})^D \tag{8}$$

where $P_0$ is the vector of parameters corresponding to our awake simulation and $P_1$ is the vector of parameters corresponding to either our anesthesia or seizure simulation. Thus, as $D$ is increased, the model's parameters move from their 'awake' values at $D = 0$ to their values in 'altered' states at $D = 1$. Moreover, reflecting biological saturation effects, the magnitude of change in model parameters becomes increasingly small as $D$ is further increased, and no parameters change signs with higher values of $D$.

## Calculating stochastic Lyapunov exponents

To determine the chaoticity of the mean-field model's dynamics, we estimated the stochastic largest Lyapunov exponent across our simulated cortical and thalamic LFPs. In general, Lyapunov exponents measure the rate of divergence between initially nearby points in a system's phase space: a positive

largest Lyapunov exponent signifies chaos (because it indicates that initially similar states diverge exponentially fast), a negative largest Lyapunov exponent signifies periodicity (because it indicates that initially similar states *converge* exponentially fast), and a largest Lyapunov exponent of zero indicates edge-of-chaos criticality, with near-zero exponents indicating near-critical dynamics (*Ovchinnikov et al., 2020*). For any given parameter configuration, stochastic Lyapunov exponents were estimated by running the model once for 20 s with random initial conditions, and then running it again, but adding a tiny random perturbation to all neural populations at 9.999 s, and then measuring the rate of the divergence of the simulated cortical and thalamic LFPs over the two runs over the final 10 s of the simulation. The divergence $\epsilon(t)$ between the first run $Q_e^{(1)}$ and the second run $Q_e^{(2)}$ was estimated as their summed squared-difference:

$$\epsilon(t) = (Q_e^{(1)}(t) - Q_e^{(2)}(t))^2 / \epsilon^{\text{max}}$$ (9)

where $\epsilon^{\text{max}}$ is the maximum possible difference between the two simulations:

$$\epsilon^{\text{max}} = \left( \max(Q_e^{(1)}) - \min(Q_e^{(2)}) \right)^2$$ (10)

The largest Lyapunov exponent $\Lambda$ of the model's dynamics is then determined by estimating the rate of divergence between the two runs $\epsilon(t)$:

$$\epsilon(t) = \epsilon(0)\exp(\Lambda t)$$ (11)

where $\epsilon(0)$ is the distance between $Q_e^{(1)}$ and $Q_e^{(2)}$ at $t = 0$. The slope of $\ln \epsilon(t)$-versus- $t$ therefore gives the estimate of the largest Lyapunov exponent. For all parameter configurations, $Q_e^{(1)}$ and $Q_e^{(2)}$ were run with identical noise inputs, meaning that the slope of $\ln \epsilon(t)$-versus- $t$ gives the stochastic Lyapunov exponent of the model.

## Human essential tremor patient propofol data

Data previously published by *Malekmohammadi et al., 2019* were re-analyzed in order to assess the relationship between the stability of neural electrodynamics and the breakdown of thalamo-cortical communication during GABAergic anesthesia. Data were collected from 10 ET patients (6 female and 4 male, ages 60–79 years) undergoing unilateral (n=6) or bilateral (n=4) implantation of deep brain stimulation (DBS) leads in the ventral intermediate (ViM) nucleus of the thalamus. All subjects provided written informed consent to participate in the original study, which was approved by the institutional review board of the University of California, Los Angeles. LFPs were recorded from the ViM thalamus, and electrocorticography (ECoG) signals were recorded from ipsilateral frontoparietal cortex during resting wake states and after intravenous propofol administration. Signals were acquired using BCI2000 v3 connected to an amplifier (g.Tec, g.USBamp 2.0) at a sampling rate of 2400 Hz. Data were bandpass filtered online between 0.1 and 1000 Hz. Patients were awake with eyes open for the first minute of recording. We used this minute of data for each patient's 'awake' state. After this first minute, the attending anaesthesiologist administered propofol intravenously. All patients reached a modified observer's assessment of alertness/sedation scale (MOAA/S) of 0, indicating no responsiveness, or 1, indicating only responses to noxious stimuli. On average, LFP and ECoG recording continued for 5 min after propofol administration. To control for cross-patient differences in blood volume, cardiac output, and propofol dosing, we exclusively analyzed the final minute of recording as each patient's 'anesthetized' state, during which they were maximally anesthetized. Data were split into 10 s trials, demeaned, detrended, and band-stop filtered at 60 Hz and harmonics (to filter out line noise). Data were then visually inspected for artifacts, and 10 s trials with artifacts spanning multiple channels were removed.

## Long-Evans rat propofol data

Data previously published by *Reed and Plourde, 2015* were re-analyzed to evaluate the effect of propofol on neural criticality and cortical-thalamic information transfer in nine male Long-Evans rats, which were included so as to rule out the possibility that our observed results in the human ET patients were driven by their pathology. Bipolar electrodes were inserted into the ventral posteromedial nucleus of the thalamus and sensory (barrel) cortex. A reference electrode was placed in the contralateral

parietal bone and a ground was placed in the ipsilateral frontal bone. Propofol was administered in the right jugular vein catheter to achieve incrementally higher plasma propofol concentrations of 3 µg/ml, 6 µg/ml, 9 µg/ml, and 12 µg/ml. Target plasma concentrations were achieved using using pharmacokinetic parameters derived from *Knibbe et al., 2005* with the Harvard-22 syringe pump, which was controlled by the Stanpump software (Department of Anesthesiology, Standford University, CA). LFPs for each condition were recorded after 15 min of drug equilibration. Unconsciousness, defined as complete loss of the righting reflex, was achieved by 9 µg/ml in all animals. In our primary analyses, we used LFPs from the 12 µg/ml condition. Data were split into 10 s trials, demeaned, detrended, and band-stop filtered at 60 Hz and harmonics (to filter out line noise). Data were then visually inspected for artifacts, and 10 s trials with artifacts spanning multiple channels were removed.

## GAERS rat seizure data

Previously published (*Miyamoto et al., 2019*) data from seven Genetic Absence Epilepsy Rat from Strasbourg (GAERS) animals (both sexes, over 16 weeks of age), which experience spontaneous 6–8 Hz generalized spike-and-wave seizures, were provided by H.M. and K.Y. and re-analyzed. Stainless steel ECoG electrodes (1.1 mm diameter) were placed over the right somatosensory cortex under 2 isoflurane anesthesia. A stainless-steel electrode, which served as both ground and reference, was placed on the cerebellum. An insulated stainless steel wire (200-µm diameter) was stereotaxically implanted in the ventroposterior thalamus contralateral to the ECoG electrode, as well as in other cortical and subcortical sites not analyzed here. For our analyses, we only selected data from generalized spike-and-wave seizures which continued for a minimum of 10 s. Data were split into 10 s trials, demeaned, detrended, and band-stop filtered at 50 Hz and harmonics (to filter out line noise). Data were then visually inspected for artifacts, and 10 s trials with artifacts spanning multiple channels were removed. Data were separated into epileptic and non-epileptic periods through careful visual inspection of LFP traces.

## C57BL/6 mouse 5-MeO-DMT data

Previously published (*Riga et al., 2018*) LFP recordings from five male, 9–16 week-old C57BL/6 mice (wild-type) following administration of either saline or 5-MeO-DMT were provided by M.S.R. and L.L.P. and re-analyzed here. For electrode implantation, animals were first pretreated with 0.05 mg/kg s.c of the analgesic buprenorphine. Thirty minutes later, anesthetic unconsciousness was induced with 2.5% isoflurane and maintained with 1.5 isoflurane. Three stabilizer screws and a ground screw were implanted, and Plastics One electrodes (Virgina, USA) were placed in medial prefrontal cortex (mPFC) and mediodorsal nucleus of the thalamus (MD), as well as other cortical areas not analyzed here (as they are not directly connected to the MD nucleus). A prophylactic antibiotic (Enrofloxacina 7.5 mg/kg s.c.) and the analgesic buprenorphine (0.05 mg/kg s.c.) were administered for 2–3 days after surgery. LFP recordings from mPFC and MD were collected at a sampling rate of 3,200 Hz using a digital Lynx system and Cheetah software (Neuralynx, Montana, USA) in a 40x40 cm open field, and bandpass filtered between 0.1 and 100 Hz. On the recording day, first 10 ml/kg saline was injected subcutaneously, and 30 min later, saline +5-MeO-DMT (5 mg/kg) was injected subcutaneously. This dose was determined based on the (previously published) finding that 1 mg/kg 5-MeO-DMT in wild-type C57BL/6 mice is sufficient to induce head-twitch responses (*Riga et al., 2016*). LFPs were recorded for 30 min for each condition. The first 5 min after each injection were excluded from the analysis, in light of prior pharmacokinetic and behavioral studies on 5-MeO-DMT in mice (*Halberstadt et al., 2011*; *Shen et al., 2011*; *van den Buuse et al., 2011*). Data were split into 10 s trials, demeaned, detrended, and band-stop filtered at 50 Hz and harmonics (to filter out line noise). Data were then visually inspected for artifacts, and 10 s trials with artifacts spanning multiple channels were removed.

## Estimating chaoticity of neural electrodynamics

To estimate the chaoticity of real low-frequency neural electrodynamics, we used the modified 0–1 chaos test. The 0–1 test for chaos was initially developed by *Gottwald and Melbourne, 2004*, who later modified the test so that it was more robust to measurement noise (*Gottwald and Melbourne, 2005*). Dawes and Freeland modified the test further, so that it could more accurately distinguish between chaotic dynamics on the one hand, and strange non-chaotic dynamics on the other (*Dawes*

and Freeland, 2008). This final modified 0–1 test involves taking a univariate time-series $\Phi$, and using it to drive the following two-dimensional system:

$$
\begin{aligned}
p(n+1) &= p(n) + \phi(n)\cos cn \\
q(n+1) &= q(n) + \phi(n)\sin cn
\end{aligned}
\tag{12}
$$

where $c$ is a random value bounded between 0 and $2\pi$. For a given $c$, the solution to *Equation 12* yields:

$$
\begin{aligned}
p_c(n) &= \sum_{j=1}^{n} \phi(j)\cos jc \\
q_c(n) &= \sum_{j=1}^{n} \phi(j)\sin jc
\end{aligned}
\tag{13}
$$

If the time-series $\Phi$ is generated by a periodic system, the motion of p and q is bounded, whereas if $\Phi$ is generated by a chaotic system, p and q display asymptotic Brownian motion. This can be quantified by assessing the growth rate of the time-averaged mean square displacement of p and q, plus a noise term $\eta_n$ proposed by *Dawes and Freeland, 2008*:

$$
M_c(n) = \frac{1}{N}\sum_{j=1}^{N}\left([p_c(j+n)-p_c(j)]^2 + [q_c(j+n)-q_c(j)]^2\right) + \sigma\eta_n.
\tag{14}
$$

where $\eta_n$ is a uniformly distributed random variable between $[-\frac{1}{2}, \frac{1}{2}]$ and $\sigma$ is the noise level. The growth rate of the mean squared displacement can be assessed using a correlation coefficient:

$$
K_c = \mathrm{corr}(n, M_c(n))
\tag{15}
$$

$K$ is computed for 100 unique values of $c$ sampled randomly between 0 and $2\pi$. The final K-statistic is the median $K$ across all values of $c$. The K-statistic will approach 1 for chaotic systems and will approach 0 for periodic systems (*Gottwald and Melbourne, 2004*; *Gottwald and Melbourne, 2005*; *Gottwald and Melbourne, 2009*; *Gottwald and Melbourne, 2008*; *Dawes and Freeland, 2008*; *Toker et al., 2020*). Finally, note that the modified test includes a parameter $\sigma$, which controls the level of added noise in *Equation 14*. Based on our prior work examining the effects of different values of $\sigma$ on the test's classification performance (*Toker et al., 2020*), we set $\sigma = 0.5$.

The 0–1 chaos test is designed to estimate chaoticity from low-noise signals recorded from predominantly deterministic, discrete-time systems. As such, steps must generally be taken to reduce measurement noise as much as possible, to determine that a signal is *not* generated by a predominantly stochastic system, and to discretize in time potentially oversampled signals from continuous time systems. Following our prior work (*Toker et al., 2022*), we effectively cleaned up measurement noise by only applying the test to low-frequency components of neural electrophysiology recordings. Low-frequency activity was extracted by band-pass filtering LFPs between 1 and 13 Hz (matching the frequency range in our analysis of spectral information transfer). Band-pass filtering was performed using EEGLAB's two-way least-squares finite impulse response filter, with the filter order set to $\frac{500\mathrm{Hz}}{13\mathrm{Hz}} \cdot \frac{85}{22}$ for an attenuation of 85 dB at the higher frequency transition band of 13 Hz, following *Harris, 2022*. However, we note that in our prior work, which only investigated the chaoticity of cortical electrodynamics slower than 6 Hz, we used the Fitting Oscillations And One Over F or 'FOOOF' algorithm to identify channel-specific slow oscillation frequencies. Following (*Armand Eyebe Fouda et al., 2014*; *Toker et al., 2022*), all signals were time discretized before application of the 0–1 chaos test by taking only all local minima and maxima, where a local extremum was defined as having a prominence greater than 10% of the maximum amplitude of a given signal. For a given 10 s window of data, the estimated chaoticity of slow thalamocortical electrodynamics was set as the median of such band-pass filtered and time-discretized signals across all available cortical and thalamic channels. Finally, we used our previously described test of stochasticity (*Toker et al., 2020*; *Toker et al., 2022*) to ensure that our neural electrophysiology recordings were produced by predominantly deterministic dynamics (*Supplementary file 5*).

## Calculating directed information flow

Because neural information flow is likely frequency-multiplexed, we used a spectral measure of information transfer, which was recently developed by *Pinzuti et al., 2020*. The measure is based on transfer entropy, an information-theoretic estimate of the amount of information transferred from a source variable $X$ to an influenced variable $Y$(*Bossomaier et al., 2016*; *Wibral et al., 2013*; *Schreiber, 2000*):

$$T_{X \to Y}^{(k,\ell)}(t, u) = I\left(Y_t; X_{t-u}^{(\ell)} | Y_{t-1}^{(k)}\right) \tag{16}$$

where $Y_t$ is the state of process Y at time $t$, while $X_{t-u}^{(\ell)}$ represents the past $\ell$ states of process X up to time $t - u$, which accounts for the source-target interaction delay $u$. $Y_{t-1}^{(k)}$ represents the past $k$ states of process Y up to time $t - 1$. This formula expresses the transfer entropy as the conditional mutual information between the current state of Y and the past $\ell$ states of X (considering the interaction delay time $u$) given the past $k$ states of Y. This measure quantifies the amount of uncertainty reduced in the future values of Y by knowing the past values of X, given the past values of Y. The delay time $u$ and the history lengths $k$ and $\ell$ can be optimized based on the specific characteristics of the processes X and Y (*Bossomaier et al., 2016*; *Wibral et al., 2013*).

In our calculation of transfer entropy, we used the Java Information Dynamics Toolkit (JIDT) (*Lizier, 2014*) to implement the method of Kraskov and colleagues (*Kraskov et al., 2004*) for model-free kernel estimation of probability distributions, which uses Kozachenko–Leonenko estimators of log-probabilities via nearest-neighbor counting (*Kozachenko and Leonenko, 1987*). We used a fixed number K=4 of nearest neighbors. We used the Ragwitz criterion (*Ragwitz and Kantz, 2002*; implemented in JIDT) to automatically determine k=1 and $\ell$=1 for all of our datasets (though we note that, in theory if not in practice, true information transfer equals transfer entropy in the long-history limit of $k \to \infty$ (*Bossomaier et al., 2016*) and so, with k=1 it is possible that we are underestimating the amount of information actively stored in the history of target variables and thereby overestimating information transfer *Wibral et al., 2013*). For the interaction delay $u$, we scanned from 0.002ms (one time-step at a sampling rate of 500 Hz) to 40ms (20 time-steps at a sampling rate of 500 Hz) and picked a value for each individual time-series pair that maximized the estimated transfer entropy between those time-series (following *Wollstadt et al., 2017*; *Wibral et al., 2013*).

The innovation described by Pinzuti and colleagues, which enables the estimation of information transfer at particular sending and receiving frequency bands, is to use the invertible maximum overlap discrete wavelet transform (MODWT) to create surrogate data in which dynamics in either the sending or receiving signal are randomized (in our case, using the Iterative Amplitude Adjustment Fourier Transform) only within a particular frequency range. The use of such surrogate signals allows both for the estimation of the *strength* of spectrally resolved information transfer (by assessing, on average, how much transfer entropy is lost when dynamics in a certain frequency range of the sender and receiver are randomized), as well as the *statistical significance* of spectral information transfer (by quantifying the percentage of surrogates which result in estimated transfer entropy greater than the estimated transfer entropy between the original sender and receiver signals).

As described by Pinzuti and colleagues, this approach can be used to determine which frequency bands are significant channels for the sending *or* receiving of information. They moreover describe a variant of their approach, which they title the 'swap-out swap-out' or SOSO algorithm, which enables the determination of the specific frequency bands from which information is sent from one channel and the frequency bands from which that same information is then received by the other channel. We used this algorithm in all spectral analyses of information transfer in this paper. In order to maximize the overlap of the frequency bands assessed by the SOSO algorithm (which are determined by successive halves of the sampling rate) with those corresponding to canonical neural oscillations, we resampled all data for our information transfer analyses to a sampling frequency of 416 Hz. In our initial exploratory analysis in *Figure 2*, we used the SOSO algorithm with only 10 surrogates (which is insufficient for determination of statistical significance) to estimate the strength of information transfer from and to all possible pairs of frequency bands between the cortex and thalamus during waking states. In subsequent analyses, we employed 100 surrogates, a number that is sufficient for the determination of statistical significance, and which additionally provides more reliable estimates of the strength of spectrally resolved information transfer. We note that we did not include the

original data in our surrogate data distributions, a method that can be utilized to provide a more conservative statistical estimate. However, to verify the robustness of our findings, we conducted an additional analysis using 250 surrogates and found the results were effectively unchanged (*Supplementary file 1*). This suggests that our findings are not substantially influenced by the number of surrogates used.

## Additional information

### Funding

| Funder | Grant reference number | Author |
| --- | --- | --- |
| National Institutes of Health | 5R01GM135420-04 | Nader Pouratian |
| Tiny Blue Dot Foundation | | Martin M Monti |

The funders had no role in study design, data collection and interpretation, or the decision to submit the work for publication.

### Author contributions

Daniel Toker, Conceptualization, Data curation, Software, Formal analysis, Funding acquisition, Investigation, Visualization, Methodology, Writing – original draft, Writing – review and editing; Eli Müller, Conceptualization, Software, Writing – review and editing; Hiroyuki Miyamoto, Maurizio S Riga, Data curation, Investigation, Writing – review and editing; Laia Lladó-Pelfort, Data curation, Investigation; Kazuhiro Yamakawa, Francesc Artigas, Data curation, Supervision, Investigation; James M Shine, Supervision, Funding acquisition; Andrew E Hudson, Conceptualization, Supervision, Funding acquisition, Investigation, Methodology, Project administration, Writing – review and editing; Nader Pouratian, Conceptualization, Data curation, Supervision, Funding acquisition, Investigation, Methodology, Project administration, Writing – review and editing; Martin M Monti, Conceptualization, Supervision, Funding acquisition, Methodology, Project administration, Writing – review and editing

### Author ORCIDs

Daniel Toker ⓘ https://orcid.org/0000-0003-0983-8937
Laia Lladó-Pelfort ⓘ http://orcid.org/0000-0003-1866-5118
Kazuhiro Yamakawa ⓘ http://orcid.org/0000-0002-1478-4390
Francesc Artigas ⓘ http://orcid.org/0000-0002-5880-5720
James M Shine ⓘ http://orcid.org/0000-0003-1762-5499

### Ethics

Ten subjects with essential tremor undergoing surgery for implantation of deep brain stimulation (DBS) leads in the ventral intermediate nucleus of the thalamus, provided written informed consent according to the Declaration of Helsinki. The institutional review board of the University of California, Los Angeles approved the study protocol.

Animal data from previously published studies were re-analyzed in this paper. The following ethics statements are quoted from the relevant papers: GAERS rats (from Miyamoto et al, 2019): "All animal experimental protocols were approved by the Animal Experiment Committee of the RIKEN Center for Brain Science. Mice and rats were handled in accordance with the guidelines of the RIKEN Center for Brain Science Animal Experiment Committee". C57BL/6 mice (from Riga et al 2018): "Animal care followed the European Union regulations (directive 2010/63 of 22/09/2010) and was approved by the Institutional Animal Care and Use Committee". Long-Evans rats (from Reed and Plourde 2015): "This study was carried out in strict accordance with the guidelines of the Canadian Council on Animal Care. The protocol was approved by the Montreal Neurological Institute Animal Care Committee. All surgery was performed under general anesthesia with ketamine and xylazine. All efforts were made to minimize suffering".

### Decision letter and Author response
Decision letter https://doi.org/10.7554/eLife.86547.sa1
Author response https://doi.org/10.7554/eLife.86547.sa2

## Additional files

### Supplementary files

- Supplementary file 1. Extended analysis of cross-frequency thalamic-cortical information transfer during sample waking-state trials, using a larger number (250) of surrogates.
- Supplementary file 2. ANCOVA results showing that brain state, but not spectral power, significantly explains the observed variances in cross-frequency thalamic-cortical information transfer.
- Supplementary file 3. Parameters for the three states (waking, anesthesia, and spike-and-wave seizure) of the mean-field model.
- Supplementary file 4. ANCOVA results showing that brain state, but not spectral power, significantly explains the observed variances in the chaoticity of low-frequency thalamocortical electrodynamics.
- Supplementary file 5. Analysis of the stochasticity of both low-frequency and broadband thalamocortical electrodynamics, suggesting predominantly deterministic behavior.
- MDAR checklist

### Data availability

The source data underlying *Figures 2–5, 8 and 9*, and code necessary to run the mean-field simulations of waking, seizure, and anesthesia states are available at figshare. The raw electrophysiology recordings from Long-Evans rats are available at the Harvard Dataverse.

The following dataset was generated:

| Author(s) | Year | Dataset title | Dataset URL | Database and Identifier |
|---|---|---|---|---|
| Toker D | 2023 | Criticality supports cross-frequency cortical-thalamic information transfer during conscious states | https://doi.org/10.6084/m9.figshare.24777081.v2 | figshare, 10.6084/m9.figshare.24777081.v2 |

The following previously published dataset was used:

| Author(s) | Year | Dataset title | Dataset URL | Database and Identifier |
|---|---|---|---|---|
| Reed SJ, Plourde G | 2015 | Data for ATTENUATION OF HIGH-FREQUENCY (50-200 HZ) THALAMOCORTICAL EEG RHYTHMS BY PROPOFOL IN RATS IS MORE PRONOUNCED FOR THE THALAMUS THAN FOR THE CORTEX | https://doi.org/10.7910/DVN/29366 | Harvard Dataverse, 10.7910/DVN/29366 |

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
