## [Editor Report]

This important study investigates thalamocortical communication and cross-frequency coupling in human and animal models under anesthesia, seizures, and the effects of the serotonergic psychedelic compound 5-MeO-DMT. These findings are exciting and compelling because they put different perturbations of brain functions – anesthesia, seizures, and psychedelic stimulation – into a single modeling framework demonstrating how these opposing perturbations reduce and enhance thalamocortical communication at specific frequencies. The evidence is compelling because it comes from multiple animal models and also incorporates a state-of-the-art neural mass model to investigate critical brain dynamics.

---

## [Decision Letter]

**Decision letter after peer review:**

Thank you for submitting your article "Criticality supports cross-frequency cortical-thalamic information transfer during conscious states" for consideration by *eLife*. Your article has been reviewed by 2 peer reviewers, and the evaluation has been overseen by a Reviewing Editor and Timothy Behrens as the Senior Editor. The following individual involved in the review of your submission has agreed to reveal their identity: Michael Wibral (Reviewer #2).

Essential revisions:

The manuscript is exciting but there are several technical issues that need to be addressed:

Both reviewers make suggestions for improving statistical analyses. These are most important to address.

It is also important to place the work in proper context by discussing prior work on neural mass models and the impact of a particular choice of dosage for the psychedelic.

*Reviewer #1 (Recommendations for the authors):*

My recommendations to improve this paper are the following:

1. Is there any behavioral information that could indicate the onset of psychedelic-like effects in the animal model? (e.g. head twitch response). Was the serum concentration of 5-MeO-DMT measured? Reporting this information could be useful to understand the effects of the dose of 5-MeO-DMT received by the animals.

2. Discuss in detail the novelty of the approach over previous studies, beyond the investigation of the 5-MeO-DMT condition, focusing on the model implementation and commonalities/differences.

3. Can channels be identified and excluded based on the presence of abnormal background activity indicative of seizures and/or tremors? Perhaps the authors could establish that the selected data did not present abnormalities that could, by themselves, drive some of the results.

4. Either perform the correction or justify why this is not needed. et al.

*Reviewer #2 (Recommendations for the authors):*

I would like to focus here on methodological and technical issues, as there are some of these that prevent a final interpretation of the results. To be clear, I do not strongly expect the main results to change dramatically, yet the way the novel Pinzuti et al. method is used here is not up to best practices and could indeed set an unfortunate example for future studies. Such difficulties in using a brand-new approach are fully understandable but nevertheless need to be corrected

1. Formula 16 on page 22: The considered variables of the random process X need not necessarily start at t-1; due to physical delays they may be found further in the past, i.e a t-δ, and then stretch to t-L. This δ can (and should) be optimized – also according to the literature that the authors cite. Also, the variables considered for the past of the random process Y need not extend to the same temporal depth L – more, or less, random variables may be the optimal choice here. From what the authors write further down, I deduce that they actually did optimize at least the delay δ; it's just that formula 16 does not properly reflect this.

2. The 'Schreiber history length' is given as k=1. There are two issues here: One minor issue is that this history length is not linked back to formula 16, thus the reader does not know which parameter in the above formula is chosen here. The second, and way more important issue is that a history length of k=1 is almost never a reasonable choice. For reasons explained for example in Wibral et al.. PLOS One, 2013 this choice of history length typically strongly underestimates the information already present in the history of the target random process (Y in formula 16); this leads to an overestimation of the transfer entropy – as explained also in detail and graphically in Lindner et al., BMC Neuroscience, 2011. Available Transfer entropy toolboxes (including Lizier's jidt, if I am not mistaken) offer ways to recursively, and automatically determine the variables that have to be considered. Also, in the original publication of Pinzuti et al., one prerequisite of using the spectrally-resolved TE is to already have an established set of the relevant variables in the source and the target random process (and not to just set k=1). I would suggest rerunning the analysis with an adapted history length. (Question: Did the authors potentially mean the Theiler exclusion length parameter (kth in the idtxl java code, I think)? )

4. What was the number of nearest neighbours K used in the analysis?

5. For all statistical tests involving Pinzuti's method it would be good to actually show the obtained surrogate-data based null distributions.

6. To me it is unclear how the initial exploratory analysis and the confirmatory statistical analysis relate. If these analyses were done on the same data, with the second analysis using a statistical test on a feature selected from the exploratory analysis of the same data with the same question (as manuscript lines 150-153 imply), then we have a clear case of so-called "double-dipping", or a circular analysis. For an explanation of this problem see Kriegeskorte et al., Circular analysis in systems neuroscience: the dangers of double dipping, Nature Neuroscience, 2009. Double dipping is considered not permissible in statistical data analysis. One solution would be to split the data – determine the feature (here: the frequency combination of interest) on a (small) subset of the data, and then run a confirmatory analysis on the remaining data. Another option would be to forego statistical testing of the cross-frequency TE and to just test and report the modulation of the spectral TE at the exploratorily chosen frequency combination by the experimental conditions. This way, the claim of having 'found' a specific and highly conserved frequency combination for thalamocortical communication would have to be dropped, but the claim to have found a modulation of frequency-specific information transfer could be upheld. A third possibility would be to not do an exploratory pre-analysis but to directly analyse the data for significant spectrally-specific TE across all relevant (see explanation below) frequency combinations, including a correction for multiple comparisons (multiple testing) performed in that case. All three possibilities for fixing this issue would be acceptable to me.

If there is a misunderstanding of the exploratory data analysis and the data used therein, please explain.

(Explanation of the use of 'relevant frequency combinations', above:) The authors claim that prohibitive computational cost made the direct statistical analysis of all frequency combinations impossible, as the number of combinations is quadratic in the number of frequencies. This statement is correct, but such an analysis is actually not necessary at all. Rather, it would suffice to first only scan the possible source frequencies for significant senders, and the target frequencies for significant receivers separately – as it is done in the original publication of Pinzuti. This problem is linear in the number of frequencies and appears tractable. After this step, only the combination of the significant source and target frequencies must be investigated with the SOSO test, again likely a low number. (Following Pinzuti et al.'s recommendation strictly, it would be only necessary to apply the SOSO test to the combination of the most significant source and the most significant target frequency, as the presence of multiple source and target frequencies leads to an assignment problem of the partial information decomposition type.; but in practice, it should be OK.)

7. If the statistical procedures were implemented by the authors themselves it would be good to know whether the original data were included once as one realization in the surrogate-data based distribution. This is good practice to mitigate the detrimental effects of two little surrogate data (like 100 used here).

8. Also, in my opinion just using 100 surrogate data for the randomiaztion test is very much on the low end of the permissible spectrum. I would much rather like to see 250+ surrogate data sets. Maybe the authors could rerun one of their most important analyses with a (much) higher number of surrogates?

9. Introduction: There are different types of critical points in neural dynamics like transitions from order to chaos, or from stable to runaway activity. Both types of transitions have received a lot of attention in neuroscience, with the first one possibly being more important for cognitive processing, while the latter seems to play a role in relation to epilepsy. It would be good if the authors made it very clear in their introduction that both types of critical transitions are topics in neuroscience and that they focus exclusively on the order-to-chaos transition. This will prevent misunderstandings.

---

## [Author Response]

Reviewer #1 (Recommendations for the authors):My recommendations to improve this paper are the following:1. Is there any behavioral information that could indicate the onset of psychedelic-like effects in the animal model? (e.g. head twitch response). Was the serum concentration of 5-MeO-DMT measured? Reporting this information could be useful to understand the effects of the dose of 5-MeO-DMT received by the animals.

The dosing of 5 mg/kg was determined based on a previous study conducted by co-authors Maurizio S. Riga and Francesc Artigas: Riga, M.S. et al. (2016), “The serotonergic hallucinogen 5-methoxy-N,N-dimethyltryptamine disrupts cortical activity in a regionally-selective manner via 5-HT1A and 5-HT2A receptors,” *Neuropharmacology.* In this previous study, it was found that just 1 mg/kg of 5-MeO-DMT was sufficient to induce a head twitch response (Figure 3B of this prior paper). We have now clarified this point in our manuscript (lines 634-636).

2. Discuss in detail the novelty of the approach over previous studies, beyond the investigation of the 5-MeO-DMT condition, focusing on the model implementation and commonalities/differences.

We have now done so in the second paragraph of our Discussion.

3. Can channels be identified and excluded based on the presence of abnormal background activity indicative of seizures and/or tremors? Perhaps the authors could establish that the selected data did not present abnormalities that could, by themselves, drive some of the results.

We appreciate your comment and the opportunity to clarify our methods regarding the exclusion of abnormal background activity indicative of seizures and/or tremors. We agree that it is important to establish that the selected data did not present abnormalities that could, by themselves, drive some of the results.

For our human essential tremor patients who were undergoing anesthesia for surgery, we ensured that the observed results were not a consequence of their pathological brain activity by also analyzing brain activity in normal rats who were given anesthesia. The inclusion of this cohort allowed us to corroborate that the results from the human essential tremor patients were not exclusively due to their specific neurological conditions but were observable under healthy conditions as well.

Regarding the GAERS rats utilized in our study, we took special precautions to visually inspect the local field potential recordings and strictly delineate between spike-and-wave seizures and non-epileptic states. This careful separation was crucial to ensure that the "waking state" we refer to in the study for this cohort truly represents periods of non-epileptic activity, hence eliminating potential confounds from pathological brain activity.

We have now clarified these points in the manuscript (lines 133-141 and 179-181) to provide a clear explanation of how we accounted for potential abnormalities in our data selection process.

4. Either perform the correction or justify why this is not needed.

Thank you for your valuable comments and your specific query regarding the application of correction for multiple comparisons in our analysis.

We understand the importance of controlling the false positive rate, especially in situations where multiple independent tests are performed. However, in the first branch of our study, we are testing a single overarching hypothesis about the presence of a statistically significant channel of communication over a specific frequency channel between the thalamus and cortex across different mammals. The individual p-values that we have reported for each subject/animal are therefore not independent hypotheses that are being tested separately. Rather, they contribute to a collective evidence base for our primary hypothesis. As such, we are not strictly conducting "multiple comparisons" in the traditional sense, where each individual comparison tests an independent hypothesis. We also note that within each subject/animal, we have accounted for multiple comparisons: we calculated cross-frequency information transfer for each 10-second window and then combined these p-values using the harmonic mean, a method designed specifically for combining dependent tests.

However, to provide further support for our conclusions, we have now conducted binomial tests on the harmonically combined p-values across all subjects/animals (lines 163-169). These tests evaluated the consistency of significant low-to-high frequency information transfer across all subjects/animals, under the assumption that the probability of observing a significant result in any given subject/animal by chance alone would be less than 0.05 (if the null hypothesis of no consistent cross-frequency information transfer were true). The binomial tests returned a p-value of 0 for both cortico-thalamic and thalamo-cortical cross-frequency information transfer during conscious states, further substantiating our primary hypothesis.

For our secondary empirical analyses, we tested specific sub-hypotheses concerning the stability of slow thalamocortical electrodynamics and the strength of cross-frequency information transfer in multiple contexts, which were directly derived from our primary hypothesis and inspired by existing literature (see our Introduction). While the tests were distinct for different contexts, they were not independent. They collectively formed the evidential base for the broader hypothesis that we were examining. To account for the risk of false positives due to multiple tests, we combined these individual tests into overarching non-parametric ANCOVA tests, yielding three "omnibus" p-values (Supplementary Files 2 and 5). This integrated approach provided a rigorous statistical analysis while mitigating the risk of inflated Type I error rate.

We acknowledge that the application of such statistical tests would be more robust if non-significant spectral channels of communication were included as controls. To address this, we have updated our Discussion section to highlight the need for future work to further explore the spectral landscape of cortical-thalamic information transfer (lines 408-416). While this study focuses on the identified significant channels, it is important to emphasize that this does not preclude the possibility of significant communication in other frequency bands. We hope that the inclusion of this acknowledgement in our Discussion section helps to contextualize our statistical approach and its results, and underscores our understanding that a more exhaustive mapping of spectral communication channels is a necessary and promising direction for future research.

Reviewer #2 (Recommendations for the authors):I would like to focus here on methodological and technical issues, as there are some of these that prevent a final interpretation of the results. To be clear, I do not strongly expect the main results to change dramatically, yet the way the novel Pinzuti et al. method is used here is not up to best practices and could indeed set an unfortunate example for future studies. Such difficulties in using a brand-new approach are fully understandable but nevertheless need to be corrected1. Formula 16 on page 22: The considered variables of the random process X need not necessarily start at t-1; due to physical delays they may be found further in the past, i.e a t-δ, and then stretch to t-L. This δ can (and should) be optimized – also according to the literature that the authors cite. Also, the variables considered for the past of the random process Y need not extend to the same temporal depth L – more, or less, random variables may be the optimal choice here. From what the authors write further down, I deduce that they actually did optimize at least the delay δ; it's just that formula 16 does not properly reflect this.

Thank you for your insightful comments and suggestions. We appreciate your attention to detail and agree that our original representation of the transfer entropy calculation in formula 16 did not adequately reflect the optimization of the delay δ and the variability in the temporal depth L for the random processes X and Y. In response to your comments, we have revised the formula for transfer entropy to better reflect these considerations. The revised formula now includes the interaction delay (which we now term *u*, following Wibral et al. 2013) and the history lengths *k* and *l* for the processes Y and X, respectively. The formula, as now written, expresses the transfer entropy as the conditional mutual information between the current state of Y and the past *l* states of X (considering the interaction delay time u) given the past *k* states of Y.

2. The 'Schreiber history length' is given as k=1. There are two issues here: One minor issue is that this history length is not linked back to formula 16, thus the reader does not know which parameter in the above formula is chosen here. The second, and way more important issue is that a history length of k=1 is almost never a reasonable choice. For reasons explained for example in Wibral et al.. PLOS One, 2013 this choice of history length typically strongly underestimates the information already present in the history of the target random process (Y in formula 16); this leads to an overestimation of the transfer entropy – as explained also in detail and graphically in Lindner et al., BMC Neuroscience, 2011. Available Transfer entropy toolboxes (including Lizier's jidt, if I am not mistaken) offer ways to recursively, and automatically determine the variables that have to be considered. Also, in the original publication of Pinzuti et al., one prerequisite of using the spectrally-resolved TE is to already have an established set of the relevant variables in the source and the target random process (and not to just set k=1). I would suggest rerunning the analysis with an adapted history length. (Question: Did the authors potentially mean the Theiler exclusion length parameter (kth in the idtxl java code, I think)? )

We appreciate your insightful comments regarding the choice of history length in our transfer entropy calculation. We understand your concerns about the potential underestimation of information present in the history of the target random process and the subsequent overestimation of transfer entropy due to our initial choice of history length, k=1.

In response to your comments, we have revisited our methodology and used the Ragwitz criterion, implemented in the Java Information Dynamics Toolkit (JIDT), to automatically determine the optimal history lengths for both the source and target variables for all of our datasets. Interestingly, the Ragwitz criterion confirmed our initial choice, determining that the optimal history length (for both the source and target variables) was indeed 1 for all datasets, without exception.

We acknowledge that this selection of parameters might be underestimating the information actively stored in the history of our time-series. We have now included a discussion of this potential limitation in our Methods section, highlighting the theoretical perspective that suggests the need for a longer history length (lines 703-707). We also now use the phrase “history length” rather than the 'Schreiber history length' in our explanation of transfer entropy calculation.

We appreciate your suggestion to rerun our analysis with an adapted history length. However, given that the Ragwitz criterion consistently determined a history length of 1 for all our datasets, we believe that our current results provide a valid representation of the dynamics of our specific datasets.

4. What was the number of nearest neighbours K used in the analysis?

We thank the reviewer for catching this omission – the number of nearest neighbors was 4, which we now clarify in our Methods section.

5. For all statistical tests involving Pinzuti's method it would be good to actually show the obtained surrogate-data based null distributions.

Unfortunately, considering the very large number of 10-second windows of data for which cross-frequency information transfer was calculated (over 3,000), it was not feasible to plot the null distributions obtained for each information transfer calculation. However, at the end of this document, we have included sample surrogate data plots for the data samples for which we used 250 surrogates in Supplementary File 1 (in response to comment 8 below).

6. To me it is unclear how the initial exploratory analysis and the confirmatory statistical analysis relate. If these analyses were done on the same data, with the second analysis using a statistical test on a feature selected from the exploratory analysis of the same data with the same question (as manuscript lines 150-153 imply), then we have a clear case of so-called "double-dipping", or a circular analysis. For an explanation of this problem see Kriegeskorte et al., Circular analysis in systems neuroscience: the dangers of double dipping, Nature Neuroscience, 2009. Double dipping is considered not permissible in statistical data analysis. One solution would be to split the data – determine the feature (here: the frequency combination of interest) on a (small) subset of the data, and then run a confirmatory analysis on the remaining data. Another option would be to forego statistical testing of the cross-frequency TE and to just test and report the modulation of the spectral TE at the exploratorily chosen frequency combination by the experimental conditions. This way, the claim of having 'found' a specific and highly conserved frequency combination for thalamocortical communication would have to be dropped, but the claim to have found a modulation of frequency-specific information transfer could be upheld. A third possibility would be to not do an exploratory pre-analysis but to directly analyse the data for significant spectrally-specific TE across all relevant (see explanation below) frequency combinations, including a correction for multiple comparisons (multiple testing) performed in that case. All three possibilities for fixing this issue would be acceptable to me.If there is a misunderstanding of the exploratory data analysis and the data used therein, please explain.(Explanation of the use of 'relevant frequency combinations', above:) The authors claim that prohibitive computational cost made the direct statistical analysis of all frequency combinations impossible, as the number of combinations is quadratic in the number of frequencies. This statement is correct, but such an analysis is actually not necessary at all. Rather, it would suffice to first only scan the possible source frequencies for significant senders, and the target frequencies for significant receivers separately – as it is done in the original publication of Pinzuti. This problem is linear in the number of frequencies and appears tractable. After this step, only the combination of the significant source and target frequencies must be investigated with the SOSO test, again likely a low number. (Following Pinzuti et al.'s recommendation strictly, it would be only necessary to apply the SOSO test to the combination of the most significant source and the most significant target frequency, as the presence of multiple source and target frequencies leads to an assignment problem of the partial information decomposition type.; but in practice, it should be OK.)

We appreciate your thorough examination of our work and raising the important issue of potential "double-dipping", as delineated by Kriegeskorte et al. in 2009. We understand your concerns about potential circularity in our analysis and acknowledge the importance of avoiding such pitfalls.

Following your suggestions and to mitigate the risk of circular analysis, we have now adopted a new approach. We split our dataset into two equal halves. The first half of all 10-second windows/trials was used in the exploratory analysis (as illustrated in Figure 2). The second half of the data was then subjected to the confirmatory statistical analysis, using 100 surrogates (presented in Table 1).

Our reanalysis shows that the results obtained using this split-data approach are nearly identical to our initial findings, reinforcing the robustness of our original conclusions. This approach, we believe, adequately avoids the double-dipping issue by ensuring that the exploratory and confirmatory analyses are performed on distinct datasets.

7. If the statistical procedures were implemented by the authors themselves it would be good to know whether the original data were included once as one realization in the surrogate-data based distribution. This is good practice to mitigate the detrimental effects of two little surrogate data (like 100 used here).

Thank you for your insightful comment. We appreciate your suggestion about including the original data as one realization in the surrogate-data based distribution. We acknowledge that this practice can help to provide a more conservative statistical estimate, particularly when using a relatively small number of surrogates, and it was not an aspect that we had considered in our initial analysis.

However, to ensure the robustness of our findings, we conducted an additional analysis using a larger number of surrogates (250), and found that our results were effectively unchanged (see the response to comment 8 below). This suggests that our findings are not substantially impacted by the number of surrogates used. Nevertheless, we agree with your point regarding the potential benefits of including the original data in the surrogate distribution. We have clarified this point in the revised methods section (lines 737-738).

8. Also, in my opinion just using 100 surrogate data for the randomiaztion test is very much on the low end of the permissible spectrum. I would much rather like to see 250+ surrogate data sets. Maybe the authors could rerun one of their most important analyses with a (much) higher number of surrogates?

Thank you for your insightful comments and suggestions. We understand and agree with your concerns about the robustness of our statistical analyses. Indeed, increasing the number of surrogates for our randomization test could potentially enhance the reliability of our results.

However, due to the large amount of data in our study and the computational cost associated with such a task, running 250+ surrogates on all the data would be highly resource-intensive and time-consuming, likely taking several months.

To address your concerns in a more feasible manner, we selected a representative sample of our data and performed an analysis using an increased number of surrogates. Specifically, we randomly picked a single 10-second window from the waking/conscious state of each patient/animal in which cross-frequency information transfer was deemed statistically significant in both directions using the initial 100 surrogates. For cases where no trials showed significant communication in both directions, the trial with the lowest combined p-value was selected (i.e., the p-value for cortico-thalamic communication plus the p-value for thalamo-cortical communication).

We then reran the cross-frequency information transfer analysis on these chosen trials using 250 surrogates. Our findings from this additional analysis, which are reported in Supplementary File 1, were consistent with our original results, confirming that there is significant bidirectional low-to-high frequency information transfer between the cortex and thalamus during conscious states.

We believe that this analysis, though not encompassing the entire dataset, provides an adequate demonstration of the robustness of our results when using a larger number of surrogates.

We hope this addresses your concerns and further supports the validity of our conclusions.

9. Introduction: There are different types of critical points in neural dynamics like transitions from order to chaos, or from stable to runaway activity. Both types of transitions have received a lot of attention in neuroscience, with the first one possibly being more important for cognitive processing, while the latter seems to play a role in relation to epilepsy. It would be good if the authors made it very clear in their introduction that both types of critical transitions are topics in neuroscience and that they focus exclusively on the order-to-chaos transition. This will prevent misunderstandings.

We appreciate your attention to the details of our work and your recommendation to clarify the types of phase transitions in neural dynamics in our introduction. In response to your comment, we have made revisions to our introduction to explicitly mention two major types of phase transitions in neuroscience, namely avalanche criticality and edge-of-chaos criticality. We have also emphasized that our study focuses exclusively on the edge-of-chaos transition, because it is likely particularly relevant for information processing (lines 78-84).